# Quantitative High-Throughput Screening Methods Designed for Identification of Bacterial Biocontrol Strains with Antifungal Properties

Bodil Kjeldgaard,[a,b] Ana Rute Neves,[a*] César Fonseca,[a] Ákos T. Kovács,[b] Patricia Domínguez-Cuevas[a]

[a]Discovery, R&D, Chr. Hansen A/S, Hoersholm, Denmark
[b]Bacterial Interactions and Evolution Group, DTU Bioengineering, Technical University of Denmark, Kongens Lyngby, Denmark

**ABSTRACT** Large screens of bacterial strain collections to identify potential biocontrol agents often are time-consuming and costly and fail to provide quantitative results. In this study, we present two quantitative and high-throughput methods to assess the inhibitory capacity of bacterial biocontrol candidates against fungal phytopathogens. One method measures the inhibitory effect of bacterial culture supernatant components on the fungal growth, while the other accounts for direct interaction between growing bacteria and the fungus by cocultivating the two organisms. The antagonistic supernatant method quantifies the culture components' antifungal activity by calculating the cumulative impact of supernatant addition relative to the growth of a nontreated fungal control, while the antagonistic cocultivation method identifies the minimal bacterial cell concentration required to inhibit fungal growth by coinoculating fungal spores with bacterial culture dilution series. Thereby, both methods provide quantitative measures of biocontrol efficiency and allow prominent fungal inhibitors to be distinguished from less effective strains. The combination of the two methods sheds light on the types of inhibition mechanisms and provides the basis for further mode-of-action studies. We demonstrate the efficacy of the methods using *Bacillus* spp. with different levels of antifungal activities as model antagonists and quantify their inhibitory potencies against classic plant pathogens.

**IMPORTANCE** Fungal phytopathogens are responsible for tremendous agricultural losses on an annual basis. While microbial biocontrol agents represent a promising solution to the problem, there is a growing need for high-throughput methods to evaluate and quantify inhibitory properties of new potential biocontrol agents for agricultural application. In this study, we present two high-throughput and quantitative fungal inhibition methods that are suitable for commercial biocontrol screening.

**KEYWORDS** *Bacillus*, *Fusarium culmorum*, *Fusarium graminearum*, *Botrytis cinerea*, fungal growth inhibition method, quantification of antifungal properties, biocontrol agents, bacterial-fungal coinoculation, bioactive compounds, biocontrol screening, bioactive metabolites, fungal growth inhibition, high-throughput screening, antifungal agents, quantitative methods

Address correspondence to Patricia Domínguez-Cuevas, dkpacu@chr-hansen.com.

*Present address: Ana Rute Neves, Arla Foods Ingredients, Sønderhøj, Denmark.

The authors declare a conflict of interest. B.K., A.R.N., C.F. and P.D.-C. are or were employed by Chr. Hansen A/S, a global supplier of biocontrol strains for the Plant Health and Nutrition sector. The remaining author (Á.T.K) declares that the research was conducted in the absence of any commercial or financial relationships that could be construed as a potential conflict of interest.

On an annual basis, it is estimated that global crop production suffers losses between 20 to 40% due to pests and plant diseases (1). Plant diseases alone are predicted to cost the global economy a staggering $220 billion per year (2). Among other plant pathogens, fungal phytopathogens contribute to considerable losses in agriculture and greatly impact food security in developing countries (3–5). Not only do fungal pathogens affect the yield, but fungal crop infections also lead to severe reductions of postharvest crop quality. For instance, the accumulation of high levels of mycotoxins renders crops unsafe for human consumption and for animal forage (6, 7). In modern intensified agriculture, fungal diseases are commonly fought using fungicides (8), but the rising fungicide resistance and chemical

pollution represent a challenge to the sustainable use of these chemicals in agriculture (9–12). In addition, many fungicides are hazardous to humans and may be implicated in developmental toxicity, reproductive defects, or cancer (13, 14). The application of microbial biocontrol agents represents a safe alternative to the intensive use of agrochemicals (11, 15). Biocontrol agents reside in close association with plant surfaces, i.e., leaves or roots, and protect the plant from phytopathogens by priming the plant defense response, competing for nutrients, and/or directly antagonizing the growth and development of the pathogenic intruders (16–18). Strains from the *Pseudomonas*, *Burkholderia*, *Streptomyces*, and *Bacillus* genera are well known for their antifungal capacity and for the production of a large variety of bioactive metabolites (15, 19–24). Although the inhibitory effect of specific soil bacteria is well documented and recognized, there is a lack of quantitative and high-throughput screening (HTS) procedures to identify competent biocontrol agents. Consequently, many potential biocontrol agents eventually fail to suppress plant diseases in field trials (25, 26). Classic antagonistic screens, which are referred to as dual culture, plate confrontation, or inhibition zone assays, assess the impact of the biocontrol candidates on the phytopathogen after coinoculation on solid media (25, 27). Such methods account for numerous factors, including nutrient or space competition, cell surface components, and the induced or constitutive secretion of volatile or soluble metabolites (21, 27–29). Other antagonistic assays evaluate the effect of individual inhibitory components, such as volatiles, polyketides, lipopeptides, siderophores, and lytic enzymes, including chitinases, glucanases, and proteases, on the phytopathogens' growth (21, 27). More complex antagonistic assays, such as leaf disc or seedling assays (30, 31), investigate the tripartite interaction between the biocontrol candidate, phytopathogen, and plant host, while nonantagonistic assays assess the importance of complementary inhibitory mechanisms, including niche colonization and priming of the plant immune response (27). Nevertheless, most screening systems are low throughput and provide only semiquantitative measurements of the inhibition potential against the fungus. Therefore, there is a need to develop more efficient screening methods combining quantitative measurements of antimicrobial activity with automation to increase the speed and reduce the resources required for the identification of good candidates.

Here, we describe two fungal inhibition methods to evaluate the antifungal potency of potential biocontrol agents. Both methods accommodate HTS of bacterial biocontrol candidates, allowing screening of a minimum of 1,000 strains per week, and provide accurate quantification of their inhibitory capacities. The major difference between the two methods is represented by the use of growing bacterial cells as opposed to (cell-inactive) culture supernatants. We demonstrate the efficacy of the methods using bacterial strains with different antifungal performance. Using the two novel methods, the antifungal properties of the bacteria were compared, and prominent fungal inhibitors were distinguished from less effective bacterial strains. Both methods were developed utilizing *Fusarium culmorum* as the model plant pathogen and *Bacillus* spp. as model antagonists. In addition, the methods were further applied for screening *Bacillus* spp. against other important phytopathogens, i.e., *Fusarium graminearum* and *Botrytis cinerea*, proving that our methods can readily be adjusted to other fungal species.

## RESULTS

**Coinoculation of fungal spores with bacterial dilution series facilitates quantification of inhibition potency.** The so-called dual-culture assay is among the most common screening methods to identify potent fungal inhibitors from microbial collections (19, 23, 32–34). Typically, the assay is performed by inoculating potential biocontrol agents at a fixed distance from the pathogenic fungal inoculum on a petri dish, as illustrated in Fig. 1A. Subsequently, the biocontrol agent's ability to suppress fungal growth is manually assessed by measuring the radius of the mycelial growth relative to the control or by measuring the size of the inhibition zone (35–38). However, accurate comparison and subsequent ranking of large numbers of strains is difficult with this assay due to the format of the readout. To improve the evaluation and accurate quantification of antifungal potency, we developed an HT fungal inhibition assay based on direct coinoculation of bacterial cultures and fungal plant-pathogenic spores. Fungal spores rather than mycelium were used as the initial inoculum in the assay to allow assessment of the biocontrol agent's impact on both spore germination and fungal

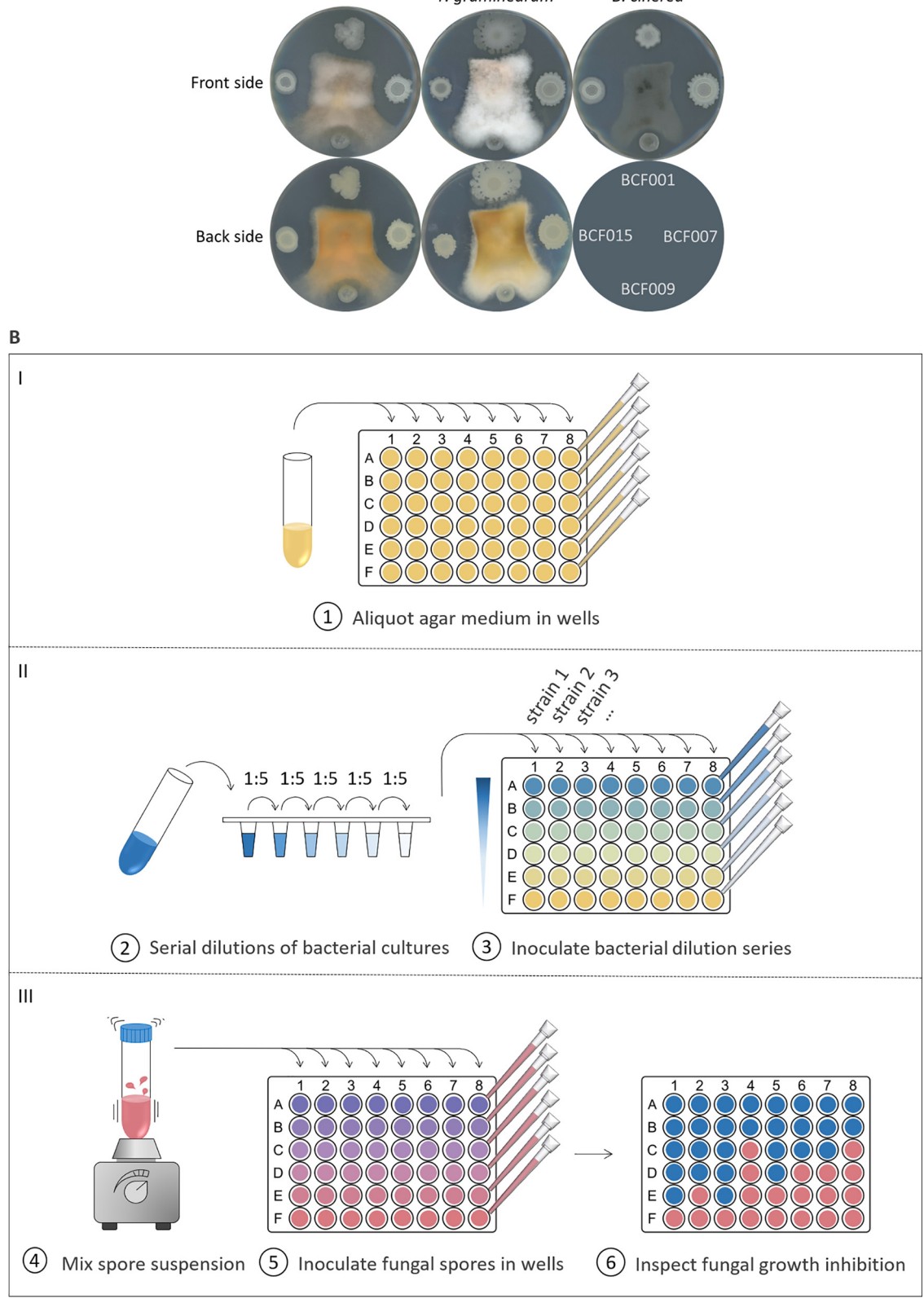

**FIG 1** Fungal inhibition assays. (A) *B. subtilis* strain BCF001, *B. amyloliquefaciens* strain BCF007, *B. paralicheniformis* strain BCF009, and *B. velezensis* strain BCF015 were spotted around central inocula of *Fusarium culmorum*, *Fusarium graminearum*, and *Botrytis cinerea* on agar medium. Inhibition zones were observed 6 days postinoculation. The backside of the *B. cinerea* plate is not shown; the scheme shown instead indicates the positions of the inoculated strains. (B) In each well of a 48-well microtiter plate, molten

**A**

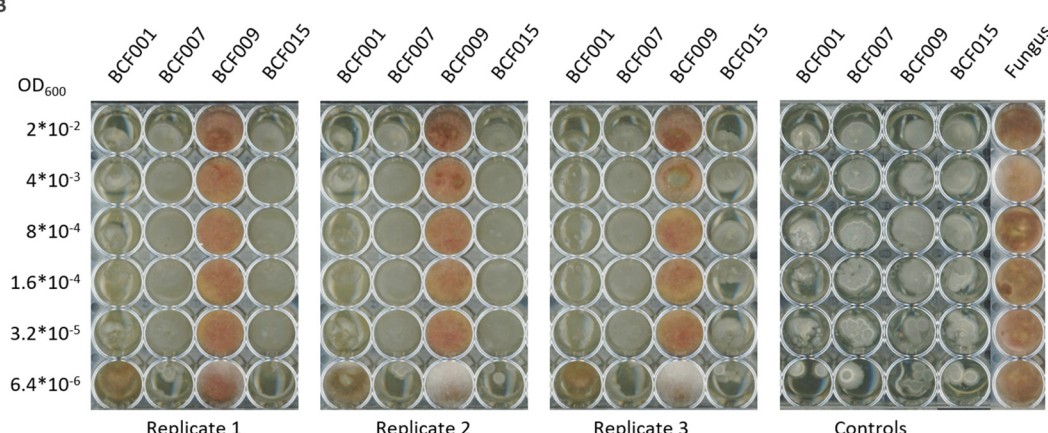

**B**

**FIG 2** Comparison of *Bacillus* species inhibitory properties against *F. culmorum*. (A) *F. culmorum* growth was scored following a three-step scale, illustrated with three example images per category. The first category shown in the top row includes wells with absence of fungal growth, where only bacterial growth can be observed; the second category shown in the middle row includes wells with signs of co-growth of both species, while the third category shown in the bottom row includes wells with only fungal growth or mainly fungal growth. (B) Dilution series of *B. subtilis* BCF001, *B. amyloliquefaciens* BCF007, *B. paralicheniformis* BCF009, and *B. velezensis* BCF015 were prepared and inoculated in consecutive columns of a 48-well microtiter plate. A constant spore concentration of *F. culmorum* was inoculated in each well. ODs of serial dilutions are indicated on the left ($OD_{600}$). The rightmost panel shows the control plate, corresponding to the growth of bacterial serial dilutions in the absence of fungal spores and to fungal growth in the absence of bacteria.

growth. First, bacterial overnight cultures were normalized to the same optical density at 600 nm ($OD_{600}$) and 5-fold serial dilutions were prepared (down to an amount of approximately 300 to 500 bacterial cells inoculated). The $OD_{600}$ of each strain used in the study was correlated with viable cell counts (CFU). A fungal spore suspension was prepared and mixed thoroughly to ensure a homogeneous spore distribution. Then, samples from the bacterial dilution series were coinoculated with a fixed quantity of fungal spores to determine the minimal inhibitory cell concentration (MICC) that abolished fungal growth (Fig. 1B). With this setup, a low MICC indicates a higher antifungal potency for a given bacterial strain. The assay was prepared on an appropriate agar medium for fungal cultivation in 48-well microtiter plates and incubated for 5 days at room temperature before assessing the fungal growth.

Four *Bacillus* strains of different species were selected based on their differential properties for fungal inhibition. The inhibitory capacities of the selected strains, *Bacillus subtilis* BCF001, *Bacillus amyloliquefaciens* BCF007, *Bacillus paralicheniformis* BCF009, and *Bacillus velezensis* BCF015, were quantified against *F. culmorum* DSM1094 using the developed method (Fig. 2B). The assay was conducted as three independent biological replicates, rendering very similar results, which indicates that the method is highly reproducible.

**FIG 1** Legend (Continued)

PDA medium was aliquoted and allowed to solidify. Fivefold dilution series of *Bacillus* strains were prepared from cultures normalized to the same $OD_{600}$. In each column, consecutive wells were coinoculated with the *Bacillus* dilution series and a fixed quantity of fungal spores (constant volume of fungal spore suspension) on the agar surface. The spore suspensions were mixed thoroughly before aliquoting. Assay results were evaluated by visual inspection after 5 days of incubation at room temperature.

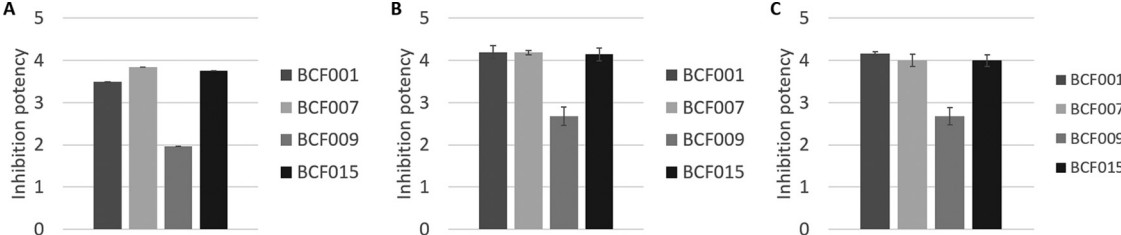

**FIG 3** Inhibition potency of *Bacillus* species against fungal phytopathogens. The inhibition potencies of *B. subtilis* BCF001, *B. amyloliquefaciens* BCF007, *B. paralicheniformis* BCF009, and *B. velezensis* BCF015 were assigned numerical scores based on the minimal inhibitory cell concentration (MICC) against *F. culmorum* (A), *F. graminearum* (B), and *B. cinerea* (C). Inhibition scores were calculated by applying equation 1 (Materials and Methods). Error bars show standard deviations.

Fungal growth inhibition was scored as no growth (1), weak growth (2), and (positive) growth (3) (Fig. 2A). Fungal growth was defined by the presence of visible hyphae in the well, including those that were half-covered by fungal mycelia. No fungal growth was defined by the complete absence of visible fungal mycelia and clear presence of bacterial growth. Weak growth was defined by the presence of barely visible hyphae in all replicates or by the absence of growth in half of the replicates. The weak growth category included the wells where *F. culmorum* produced (orange) pigmentation, even in the absence of visible hyphae. The coinoculation of bacterial culture dilutions and fungal spores was used to distinguish the limits of the bacterial inhibition capacity.

Comparison of bacterial MICCs allowed ranking of the strains in accordance with their fungal inhibition properties (Fig. 3A, Table S1 in the supplemental material). The strains *B. amyloliquefaciens* BCF007 and *B. velezensis* BCF015 showed the most potent inhibition properties, and even at the highest dilution step (corresponding to an initial $OD_{600}$ of $6.4 \times 10^{-6}$, equivalent to 340 to 500 CFU inoculated), the two strains were able to inhibit *F. culmorum*'s growth. Determining the MICC more accurately would require smaller dilution steps to be included in the assay. Nevertheless, the experimental setup is optimized for HTS of large strain collections. *B. subtilis* BCF001 also displayed potent fungal inhibition properties, abolishing fungal growth up to an $OD_{600}$ of $3.2 \times 10^{-5}$, corresponding to 1,900 CFU inoculated. *B. paralicheniformis* BCF009, however, did not affect fungal growth even at the lowest dilution step, corresponding to an $OD_{600}$ of $2 \times 10^{-2}$, or $3.89 \times 10^4$ CFU inoculated.

To generate visual and directly comparable plots of the inhibition results, we calculated a numerical inhibition score that reflects the inhibitory capacity of each strain. In brief, the lowest cell concentration that caused fungal growth inhibition was identified for each strain, and a numerical inhibition score was calculated based on the natural logarithm to the MICC by applying the empirical formula presented in equation 1 in Materials and Methods (Fig. 3A, Table S1). Plotting the inhibition scores clearly indicated that *B. amyloliquefaciens* BCF007 and *B. velezensis* BCF015 were the most efficient strains inhibiting the growth of *F. culmorum*. *B. subtilis* BCF001 also showed a high inhibition score, although smaller than the ones calculated for the two former strains. The scoring results for *B. paralicheniformis* BCF009 matched the low inhibitory activity of this strain against *F. culmorum*. The MICCs of all strains against *F. culmorum* were significantly different from each other (Table S2).

**Evaluation of the inhibition method with additional fungal phytopathogens.** Using the newly developed method, the antifungal activities of *B. subtilis* BCF001, *B. amyloliquefaciens* BCF007, *B. paralicheniformis* BCF009, and *B. velezensis* BCF015 were also quantified against other phytopathogenic filamentous fungi, specifically *F. graminearum* and *B. cinerea* strains. Images acquired to record the experimental results and the calculated MICCs and inhibition scores can be found in Fig. S1 and Table S1. Interestingly, our inhibition assay proved applicable to these additional species of plant-pathogenic fungi. The initial spore concentration was the only parameter that required adjustment when testing growth inhibition against the new fungal strains.

The above-described scoring system was applied to the results obtained for the three fungal species (Fig. 3). While *B. amyloliquefaciens* BCF007 and *B. velezensis* BCF015 displayed

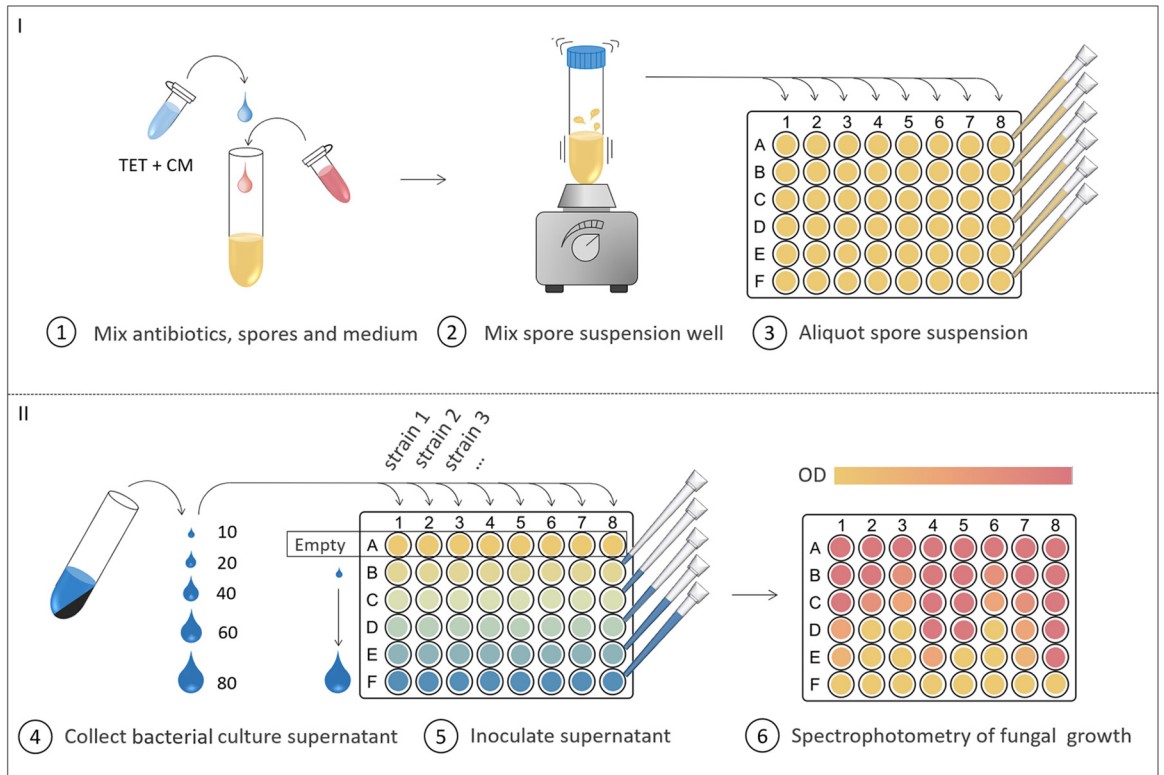

**FIG 4** Supernatant inhibition method. A fungal spore suspension was prepared with PDB medium and the bacteriostatics tetracycline (TET) and chloramphenicol (CM). The suspension was mixed well and aliquoted into each well of a 48-well microtiter plate. *Bacillus* overnight cultures were adjusted to an $OD_{600}$ of 2 and were subsequently centrifuged to collect the supernatants. In each column, a range of volumes of *Bacillus* supernatant (10 to 80 $\mu$l) were added to consecutive wells. Fungal growth was evaluated by spectrophotometric measurements after 5 days of incubation at 25°C in darkness.

the most efficient growth inhibition of *F. culmorum*, the differences between them and *B. subtilis* BCF001 were minimal when assayed against *F. graminearum* DSM4528, and the MICCs were not significantly different (Table S2). Furthermore, *B. subtilis* BCF001 proved more efficient than *B. amyloliquefaciens* BCF007 and *B. velezensis* BCF015 against *B. cinerea* Kern B2. The inhibition scoring results obtained for *B. paralicheniformis* BCF009 were in accordance with the poor inhibitory properties of this strain, regardless of the fungal pathogen tested. For comparison between strains with similar inhibition potencies, the range of the dilution series can readily be adjusted to allow determination of a more accurate MICC.

**Inoculation of fungal spores with bacterial culture supernatants validates the inhibitory importance of supernatant components.** While the antagonistic coinoculation method assesses several inhibitory mechanisms, the use of cell-inactive supernatants accounts for the bioactivity of secreted metabolites, such as enzymes, lipopeptides, and polyketides, on the fungus. In the antagonistic supernatant method, fungal growth was estimated by $OD_{600}$ measurements in liquid cultures, and therefore, reduced fungal growth in the presence of bacterial metabolites reflects greater inhibition potency of a given strain. For this method, bacterial cultures were grown overnight and normalized to the same $OD_{600}$ value prior to collection of the culture supernatants by centrifugation (Fig. 4). A range of increasing supernatant volumes (10 to 80 $\mu$l) were inoculated into fungal spore suspensions prepared in liquid medium in 48-well microtiter plates. To avoid any possibility of misreads arising from remaining bacterial cells present in the supernatants, bacteriostatic antibiotics were added to the fungal spore suspensions to prevent bacterial growth. The microtiter plates were incubated statically at 25°C in darkness. Subsequently, the fungal growth was quantified by spectrophotometric measurement. An area scan protocol (5 by 5 measurements) was used to account for empty spaces and clumps of fungal growth within the well. Optical density measurements were previously found to correlate to fungal dry weight (39–41). Dynamic

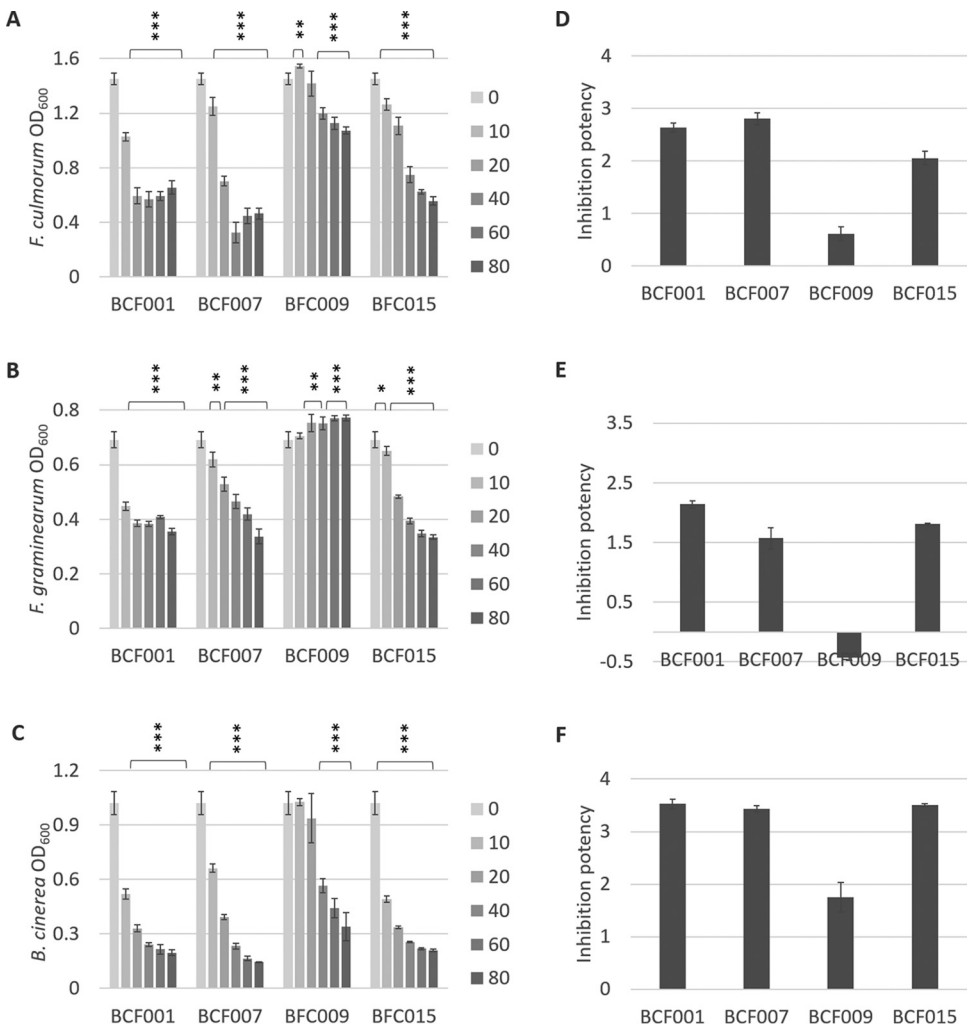

**FIG 5** Fungal growth inhibition by *Bacillus* species culture supernatants. *Bacillus* species cultures were grown overnight in LB broth and normalized to an $OD_{600}$ of 2. Different volumes of cell-inactive supernatants (10, 20, 40, 60, and 80 $\mu$l) were added to 48-well microtiter plates containing PDB medium and a fixed fungal spore concentration. Bacterial growth was inhibited by the presence of bacteriostatic antibiotics in the culture medium (50 $\mu$g/mL chloramphenicol and 10 $\mu$g/mL tetracycline). Plates were incubated at 25°C without shaking for 60 h (in darkness), and fungal growth was measured by spectrophotometry at 600 nm. Plots correspond to inhibition results for *F. culmorum* (A), *F. graminearum* (B), and *B. cinerea* (C). Statistical significance for each supernatant volume compared to the control was calculated based on biological triplicates. \*, $P < 0.05$; \*\*, $P < 0.005$; \*\*\*, $P < 0.0005$. For quantification of the culture supernatant inhibition potency, an inhibition score was assigned to each *Bacillus* strain by subtracting the accumulated relative fungal growth of *F. culmorum* (D), *F. graminearum* (E), and *B. cinerea* (F) in response to all supernatant volumes from the total potential growth according to equation 2 (Materials and Methods). Standard deviations were calculated based on biological triplicates.

assessment of fungal growth was done using this methodology, and fungal growth curves may be found in Fig. S2.

The inhibition assay demonstrated a reverse correlation between the added volume of bacterial cell-inactive supernatant and the growth of *F. culmorum*, *F. graminearum*, or *B. cinerea* (Fig. 5A to C). For the potent fungal inhibitory strains, including *B. subtilis* BCF001, *B. amyloliquefaciens* BCF007, and *B. velezensis* BCF015, increasing volumes of bacterial supernatant correlated with a progressive decrease of fungal growth. The addition of *B. paralicheniformis* BCF009 supernatant impacted fungal growth to a much lesser extent, which was in accordance with the results obtained with the coinoculation method. In some cases, a slight increase in fungal growth was observed in response to the addition of the largest supernatant volume (80 $\mu$l culture supernatant), suggesting an effect of supplemented nutrients (not consumed by the bacteria) that outweighed the effect of the antifungal bioactive metabolites.

**Scoring of supernatant inhibition potency.** A numerical inhibition score was assigned to each strain by quantifying the fungal growth in response to supernatant addition (5 different volumes). The inhibition scores were calculated by subtracting the accumulated relative fungal growth from the total potential fungal growth under the 5 conditions tested (equation 2 in Materials and Methods).

The inhibition scores can be found in Table S1. Culture supernatants of *B. subtilis* BCF001 and *B. amyloliquefaciens* BCF007 showed the most effective growth inhibition of *F. culmorum* (Fig. 5A and D). Against *F. graminearum*, *B. subtilis* BCF001 culture supernatants proved the most inhibitory, while *B. subtilis* BCF001, *B. amyloliquefaciens* BCF007, and *B. velezensis* BCF015 all displayed similar inhibition potencies against *B. cinerea*. *B. paralicheniformis* BCF009 culture supernatants inhibited *F. culmorum* growth poorly.

It is worth mentioning that assays prepared from different fungal spore stock solutions showed small variations in final fungal growth (Fig. S3), possibly due to variations in the initial spore count or spore viability. These variations could be minimized either by prolonging the incubation time until readout or by preparing assays from a unique spore stock. For this reason, biological replicates were prepared with the same fungal spore solution. Nevertheless, the overall conclusions remained unchanged despite variations in the final fungal growth.

**Comparison of methods to remove or inactivate cells from culture supernatants.** Typically, bacterial-cell-inactive supernatants are generated by filtration of the cell cultures (42–45). In contrast, in the setup proposed here, cell-inactive supernatants were generated by centrifugation to pellet bacterial cells. To avoid cell growth of remaining bacteria in suspension, we added the bacteriostatic antibiotics chloramphenicol and tetracycline. Plating of the fungal growth suspension on bacterium-selective medium after the endpoint measurement produced no bacterial growth (no colonies), indicating that the antibiotics effectively inhibited the growth of any putative remaining bacterial cells. The final fungal growth reached by *F. culmorum*, *F. graminearum*, and *B. cinerea* was somewhat affected by the addition of the respective antibiotics (Fig. S4). The results obtained with filtered supernatants with and without antibiotics revealed that the relative inhibition results remained unaltered, although the addition of antibiotics slightly but significantly reduced the fungal growth (Fig. S5, Table S3). Both filtration and antibiotic addition had generally no effect on the fungal inhibitory potency, suggesting that bioactive metabolites remained active after both procedures (Fig. S6, Table S3).

**Comparison of the two methods proposed in the present study and the dual-culture assay.** The two proposed quantitative HT methods were compared to the common dual-culture assay using plates inoculated with fungus and *Bacillus* strains (Fig. 1A). In accordance with the results from the two methods, zones of fungal growth inhibition were observed around the bacterial colonies of *B. subtilis* BCF001, *B. amyloliquefaciens* BCF007, and *B. velezensis* BCF0015, whereas the growth of *F. culmorum*, *F. graminearum*, and *B. cinerea* was nearly unaffected by *B. paralicheniformis* BCF009. The observed growth reduction from the dual-culture assay was in accordance with the calculated MICCs from the antagonistic coinoculation assay (Fig. 1A and 3, Table S1). Indeed, the traditional dual-culture assay allows the assessment of fungal inhibitory capacity and enables a quick and simple evaluation of biocontrol strains' potential, but the results lack the accurate quantification of the bacterial inhibition potency provided by the methods proposed here. By adjusting the range of the dilution series, genetically or phenotypically similar strains can be compared and ranked using the antagonistic coinoculation method. This allows the comparison of similarly potent biocontrol candidates. In addition, the simple numerical scoring systems allow easy comparison between a large number of strains. The throughput is approximately 1,000 strains per week for each of the two screening methods described, including preparation time and readout of the assays, but not taking incubation time into consideration. The cost of the assays includes the operation cost of the robot, pipette tips, growth medium, and one 48-well microtiter plate per 8 strains. Although the classical dual-culture assay is undoubtably more cost-effective, the quantification aspect of the methods described in this study constitutes a great advantage when comparing biocontrol candidates, which is valuable for commercial screening.

While the antagonistic supernatant method specifically evaluates the inhibition potency of secreted metabolites, such as enzymes, lipopeptides, and polyketides, the antagonistic

coinoculation method quantifies the inhibitory effect of actively growing bacteria and, thereby, accounts for additional factors like competition for nutrients and space. Therefore, differences in results obtained by the two methods could provide insights into the mode of action and serve as the starting point for in-depth characterization of the molecular inhibitory mechanisms.

## DISCUSSION

Despite the steadily growing market share of biocontrol products compared to the use of conventional pesticides (16, 46), plant pathogen management continues to rely heavily on chemical substances with associated risks to human health and the environment (47). Regardless of the increased (research) efforts on microbial biocontrol product development, their deployment into the market is difficult due to the absence of a harmonized global framework, public misinformation, and complex regulation and registration procedures, as well as the high costs of field trials (48–51). The application of reliable methods to identify and select potent biocontrol strains is of critical importance to reduce product development costs and accelerate their market implementation.

Large screens of strain collections are often laborious and expensive, with the cost being proportional to the complexity of the screening system (52). Simple HTS methods are cost-effective; however, they often fail to provide quantitative results for accurate comparison of the biocontrol strains. Furthermore, following the HTS and candidate selection, many strains eventually show low efficacy in field trials, demonstrating a discrepancy between *in vitro* laboratory conditions and *in planta* application experiments. To reduce costs and minimize failed tests, it is therefore crucial to identify biocontrol candidates in primary screens before moving on to complex experimental systems, such as greenhouse or field trials. Accordingly, the development of simple, reliable, and quantitative HTS methods is crucial for ranking and selection of the best biocontrol candidates.

In this study, bacterial strains were ranked according to their bioactivity against fungal plant pathogens by using two novel HTS methods: (i) an antagonistic cocultivation method based on the inoculation of a constant number of fungal spores together with bacterial dilution series on solid medium to generate a quantitative measurement of the minimal inhibitory (bacterial) cell concentration (MICC) of fungal growth and (ii) an antagonistic cell-inactive-supernatant method based on the inoculation of a constant number of fungal spores together with different volumes of bacterial culture supernatants in liquid medium to provide a quantitative measurement of fungal growth inhibition by secreted metabolites.

Although classic antagonistic methods that assess the adjacent growth of two species on agar plates allow the assessment of fungal inhibition potential (19, 32–34, 53–55), their accuracy is limited. Factors like the inoculum size, diffusion rate of metabolites, and differential conditions between agar plates contribute to the inaccuracy of classic antagonistic assays and impair the ranking of biocontrol candidates. The quantification provided by the antagonistic cocultivation method proposed here allows the ranking of biocontrol candidates according to their inhibitory strength by means of MICC values. In addition, the direct coinoculation of fungal spores with bacterial cells allows assessment not only of the impact on mycelial growth but also on fungal spore germination, contrary to the classic agar plate methods. This type of antagonistic method, employing direct coinoculation of biocontrol candidate and pathogen, has been developed for other applications; for instance, lactic acid biocontrol bacteria coinoculated with food spoilage fungi in studies of food products in the dairy industry (56, 57). However, these assays lack precise quantification of inhibition potency, regardless of the similarities to the coinoculation method described here.

Compared to simple dual-culture assays like ours, the more complex *in planta* assays include the tripartite interaction of the biocontrol candidate, phytopathogen, and plant host (30, 31, 58) and thereby attempt to mimic field settings. Despite the closer resemblance to natural conditions, the complexity of these screening systems limits the throughput (59) and leads to additional variation (60), which complicates the interpretation of results compared to our method. However, screens that consider the interaction between three biological systems

may be advantageous to apply to a subset of biocontrol candidates following an initial HTS (59, 61).

While the antagonistic cocultivation methods investigate the direct interaction between pathogen and biocontrol candidate, the antagonistic supernatant methods estimate the inhibition potency of secreted bioactive metabolites from the biocontrol candidate on the pathogen. The experimental setup can be either agar- or liquid-based, like the proposed antagonistic cell-inactive-supernatant method described in this study. Agar-based screens rely on the evaluation of fungal colony growth or inhibition zones in response to the addition of supernatants from the biocontrol candidates (34, 43, 62). Some assays allow a quantitative comparison of inhibition capacity between strains by calculating the reciprocal to the highest supernatant dilution that exhibits a clear zone of inhibition (43, 62). Even so, the scoring is notably laborious and low throughput compared to the proposed antagonistic supernatant method described in this study.

Liquid-based supernatant assays depend on the assessment of fungal morphology in response to the biocontrol supernatants by microscopy (62, 63), evaluation of fungal growth by dry weight (64), or spectrophotometry (36, 42, 55). The latter two share a great resemblance to the proposed cell-inactive-supernatant method and allow HTS and reproducible assessment of the potency of antifungal metabolites. However, the accurate quantification of antifungal capacity by determination of supernatant inhibition scores in the method proposed here allows easy benchmarking and comparison between biocontrol supernatants, which represents a major advantage over previously reported methods.

Both assays were initially developed utilizing *F. culmorum* as the model plant pathogen and different *Bacillus* species as model antagonists. Subsequently, the assays were validated with the relevant plant-pathogenic fungi *B. cinerea* and *F. graminearum*. The results demonstrated that the methods are robust and can readily be adapted for different fungal species. Moreover, the methods can be adapted to assess the potential for inhibiting bacterial plant pathogens. In agriculture, not only fungal plant pathogens but also bacterial plant pathogens contribute to significant yield losses (65). The adaptation of the proposed methods to quantify inhibition potency against plant-pathogenic bacteria simply requires a fluorophore-labeled bacterial pathogen, a bacterial pathogen with a selective marker (i.e., antibiotic resistance), or a pathogen-selective growth medium in order to make it applicable for HTS. For discrimination between bacterial strains with similar inhibition potencies, the antagonistic cocultivation method can be adjusted by using a smaller dilution factor. Furthermore, the dilution factors can be adjusted to fit more or less potent biocontrol candidates. Finally, the applicability of the methods may even be extended to other areas, such as biocontrol screens against human or animal pathogens or screens against food spoilage microorganisms.

Several previous studies employed combinatory screens with two or more assays for the identification of potent pathogen inhibitors (33, 35, 55, 58, 63, 66, 67). In this study, we propose the combination of two HT antagonistic methods to improve confident selection of potent biocontrol bacterial strains. While the antagonistic cocultivation method accounts for the entire repertoire of direct inhibitory mechanisms displayed by a biocontrol strain, such as bioactive compounds and competition for nutrients and space, the antagonistic supernatant method pinpoints the inhibitory effect of secreted metabolites, such as enzymes, lipopeptides, and polyketides. Thus, the two methods provide different results that in combination may aid further mechanistic elucidation. In addition, while the antagonistic coinoculation method may facilitate the identification of potent candidates ideal for in-furrow application or seed coating, where pathogens and biocontrol agents actively compete, the antagonistic supernatant method identifies high producers of bioactive metabolites that would be advantageous in a liquid formulation for foliar product application. In light of the results obtained and considering the different mechanistic aspects involved in pathogenic inhibition, we argue that the combination of our novel antagonistic cocultivation and supernatant methods constitutes an improved strategy for biocontrol strain identification. The combination of the two methods presented (i) confidently reflects the fungal inhibition capacity of biocontrol candidates, (ii) facilitates HTS of large strain collections, (iii)

**TABLE 1** Fungal and bacterial strains used in the present study

| Species | Strain | Source |
|---|---|---|
| Filamentous fungi | | |
| *Fusarium culmorum* | DSM1094 | DSMZ German collection |
| *Fusarium graminearum* | DSM4528 | DSMZ German collection |
| *Botrytis cinerea* | Kern B2 | University of California, Davis |
| Bacteria | | |
| *Bacillus subtilis* | BCF001 | Chr. Hansen A/S |
| *Bacillus amyloliquefaciens* | BCF007 | Chr. Hansen A/S |
| *Bacillus paralicheniformis* | BCF009 | Chr. Hansen A/S |
| *Bacillus velezensis* | BCF015 | Chr. Hansen A/S |

can provide valuable insights into types of inhibition mechanisms for further studies, and (iv) allows easy comparison of strains by accurate quantification of their inhibition potencies.

## MATERIALS AND METHODS

**Microbial species and growth conditions.** The fungal and bacterial strains used in the inhibition assays are shown in Table 1. Fungal species were cultivated on potato dextrose agar (PDA) medium (4 g/L potato infusion, 20 g/L glucose, 15 g/L agar, pH 5.6 $\pm$ 0.2; Carl Roth). For inhibition assays on PDA, cultures were incubated at room temperature with natural light. For inhibition assays in broth, fungal spores were inoculated in potato dextrose broth (PDB) medium (6.5 g/L potato infusion, 20 g/L glucose, pH 5.6 $\pm$ 0.2; Carl Roth) and incubated at 25°C in darkness. Bacilli were grown overnight in lysogeny broth (10 g/L Bacto tryptone [Difco], 5 g/L yeast extract [Oxoid], 10 g/L NaCl [Merck], pH 7.2 $\pm$ 0.2) at 37°C with agitation at 250 rpm.

**Fungal spore harvest.** Spores were harvested as described by Benoit et al. (68) and Kjeldgaard et al. (69). In brief, PDA plates were inoculated with fungi and incubated at 22°C with 16-h-light/8-h-dark cycles for at least 2 weeks. Fungal spores were harvested from PDA plates using saline Tween solution (8 g/L NaCl, 0.05 mL/L Tween 80) and gentle scraping with an L-shaped spreader. The spore solution was filtered through double-layered Miracloth (Millipore) and pelleted by centrifugation at 5,000 rpm for 10 min. The supernatant was discarded, and the spore pellet resuspended in saline Tween. Spore stocks were prepared by adding an appropriate concentration of glycerol. The spore concentration was determined by counting using a Fast-Read 102 counting chamber.

**Quantification of fungal growth inhibition by coinoculation with bacterial dilution series.** The fungal inhibition assay was prepared in 48-well microtiter plates with 0.5 mL PDA in each well. *Bacillus* cultures were adjusted to $2 \times 10^{-2}$ or $8 \times 10^{-4}$ at OD$_{600}$, and 5-fold dilution series were prepared with 6 steps using peptone saline as the diluent (maximum recovery diluent, 9.5 g/L; Millipore). Fifteen microliters of each bacterial dilution was inoculated in consecutive wells by spotting the solution in the center of each well. One strain was assigned per column. Next, a fungal spore solution was prepared with peptone saline and vortexed vigorously to disperse clumps of spores. Fifteen microliters of fungal spore solution was coinoculated with the bacterial dilutions in each well. The approximate final fungal spore concentrations of *F. culmorum*, *F. graminearum*, and *B. cinerea* were $5.5 \times 10^6$ spores/mL, $6.25 \times 10^4$ spores/mL, and $2 \times 10^7$ spores/mL, respectively. Coinoculated plates were sealed with 3M tape (0.5 cm; Millipore) to reduce growth differences between inner and peripheral wells and incubated at room temperature under natural light conditions. After 5 days, the fungal inhibition was evaluated by visual inspection and the plates were imaged by scanning (Epson Perfection V800 Photo). The minimal inhibitory cell concentration (MICC) was identified for each strain and averaged between technical duplicates and then between biological triplicates. The MICC (CFU) against each fungal species was converted to an inhibition score using equation 1, below, and statistical significance was calculated by the two-tailed Student's *t* test.

$$\text{Inhibition score} = 5 - \frac{\text{Ln(MICC)}}{5} \qquad (1)$$

**Quantification of fungal inhibition potency by metabolites in bacterial (cell-inactive) supernatants.** The fungal inhibition assay was prepared in 48-well microtiter plates with 0.5 mL PDB medium. Spores of *F. culmorum*, *F. graminearum*, or *B. cinerea* were added to a final concentration of $1.1 \times 10^6$ spores/mL, $1.25 \times 10^4$ spores/mL, or $2 \times 10^6$ spores/mL, respectively. The spore suspensions were mixed thoroughly by vortexing to ensure a homogeneous distribution. *Bacillus* species cultures were adjusted to an OD$_{600}$ of 2 and centrifuged to pellet the cells. The supernatants were collected, and the cell pellets discarded. Increasing volumes of bacterial supernatants (ranging from 10 to 80 $\mu$l) were inoculated into the fungal spore suspensions. Two approaches were implemented for sterilization of the supernatant to omit bacterial growth. Either the bacteriostatic antibiotics chloramphenicol and tetracycline were added to the fungal spore suspension to a final concentration of 50 $\mu$g/mL and 10 $\mu$g/mL, respectively, or the supernatants were sterilized by filtration (0.2-$\mu$m Minisart syringe filter; Sartorious) prior to coinoculation with the fungal spores. The plates were sealed with 3M tape and incubated at 25°C without shaking in the dark. The fungal growth was quantified by spectrophotometric measurements (OD$_{600}$) after 48 h for *F. graminearum* and *B. cinerea* inhibition assays and after 67 to 72 h

for *F. culmorum* inhibition assays. For more accurate fungal growth estimation, the $OD_{600}$ was measured using a 5-by-5 area-scanning matrix, resulting in 25 measurements per well, which were subsequently averaged. Following the evaluation of fungal growth, the supernatants were collected and plated on LB with fungicides (50 mg/L nystatin) to check for unwanted bacterial growth. Statistical analyses were done to compare the effects of (i) bacterial culture supernatants on fungal growth, (ii) filtration and antibiotics on the bacterial supernatants' potency, and (iii) prokaryotic antibiotics on fungal growth. Statistical significance was calculated by Student's *t* test (two tailed). A numerical inhibition score was assigned to each strain by quantifying the fungal growth in response to the addition of supernatant (5 volumes). The inhibition scores were calculated by subtracting the accumulated relative fungal growth from the total potential fungal growth (5 conditions tested) as shown in equation 2:

$$5 - \mathrm{SUM}\,(\text{relative fungal growth}) \qquad (2)$$

## SUPPLEMENTAL MATERIAL

Supplemental material is available online only.

**SUPPLEMENTAL FILE 1**, PDF file, 0.5 MB.

## ACKNOWLEDGMENTS

This project was supported by Chr. Hansen A/S and the Danish National Research Foundation (grant number DNRF137) for the Center for Microbial Secondary Metabolites (DTU). B.K. was funded by a grant from the Innovation Fund Denmark (grant number 8053-00109B). The funders had no role in study design, data collection and interpretation, or the decision to submit the work for publication.

B.K., Á.T.K., A.R.N., and P.D.-C. designed the research, B.K. performed the research, B.K., Á.T.K., C.F., and P.D.-C. analyzed the data, and B.K., Á.T.K., C.F., and P.D.-C. wrote the manuscript; all authors approved the manuscript.

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
