## [Reviewer comments · Microbiology Spectrum]

Microbiology Spectrum

Quantitative high-throughput screening methods designed for identification of bacterial biocontrol strains with antifungal properties

Patricia Domínguez-Cuevas, Bodil Kjeldgaard, Ana Neves, César Fonseca, and Ákos T Kovács

Corresponding Author(s): Patricia Domínguez-Cuevas, Chr. Hansen

Review Timeline:

Submission Date:	August 30, 2021
Editorial Decision:	November 10, 2021
Revision Received:	December 22, 2021
Accepted:	February 7, 2022

Editor: Lea Atanasova

Reviewer(s): Disclosure of reviewer identity is with reference to reviewer comments included in decision letter(s). The following individuals involved in review of your submission have agreed to reveal their identity: Sergio Casas-Flores (Reviewer #1); Lidia Blaszczyk (Reviewer #2)

Transaction Report:

DOI: <https://doi.org/10.1128/Spectrum.01433-21>

November 10, 2021

Dr. Patricia Domínguez-Cuevas
Chr. Hansen
Discovery, R&D
Bøge Allé 10-12
Hørsholm 2970
Denmark

Re: Spectrum01433-21 (Development of quantitative high-throughput screening methods for identification of antifungal biocontrol strains)

Dear Dr. Patricia Domínguez-Cuevas:

Thank you for submitting your manuscript to Microbiology Spectrum. After careful consideration, we think your publication has merit but does not fully meet the journal's publication criteria as it currently stands. Therefore, we invite you to submit a revised version of the manuscript that addresses the points raised during the review process.

Link Not Available

Sincerely,

Lea Atanasova

Journals Department
Reviewer comments:

Reviewer #1 (Comments for the Author):

The manuscript describes two quantitative and high-throughput methods to evaluate the inhibitory capacity of bacterial biocontrol candidates against fungal phytopathogens. The first described method uses growing bacteria cocultured with the fungus, whereas the second one shows potential to measure the inhibitory effect of bacterial culture supernatant components on the fungal growth.

Since my view, the manuscript is well written and technically sounds, however, I have some major and minor concerns depicted below:

Lines 1-2. Since my view, the title must reflect that the methods were designed for the screening of biocontrol bacterial strains.

Line 25 and subsequently. Please write down ".spp" in plane letter instead of italic.

Line 55. I am not sure if this part of the sentence is right. I read the sentence as if the biocontrol agents emerge/born from PGPR.

Line 61. What HTP means? Please describe every abbreviation or acronym you use through the text. Furthermore, I think the right abbreviation for high-throughput is "HT". Please revise it.

Line 79. Why do you think your proposed method belongs to high-throughput approaches? I do not agree with such statement. To my knowledge HT approaches are designed for testing from hundreds to thousands of samples in a short period of time.

Lines 100-101. Spores and mycelium do not have the same structure and composition of their cell walls, so, why not to include mycelium?

Figure 1A. Contrasting with the first two plates, I cannot see the fungus nor the bacteria in the third plate. Why?

Figure 2A. What should I see in each column? I can see different morphologies and growth, then, I do not know why you said that there is not fungal growth. Please explain. Why did you not include the control of the fungi and the bacteria growing alone? That could help to understand what you meant. I can see fungal growth in all nine wells with different morphology, so why you stated that there is no growth in the first line. Should I see something that you did not explain?

Figure 2B. If I apply a zoom on the different wells, I can see growth on the different lines, so I would like to see the control of the fungi and the different Bacilli growing alone under the different conditions.

Lines 145-148. I cannot see in figure 2 such dilutions and does not seem to be the initial OD as stated. Have I got confused?

Line 152. Did you mean 3.2×10^{-6} .

Figure 3A. Above, you concluded that *B. amyloliquefaciens* and *B. velezensis* showed the most potent inhibition properties, and even at the highest dilution step against *F. culmorum*. However, *B. subtilis* showed similar results. It is necessary an statistical analysis.

Lines 191-202. How did you determine the fungal growth by OD In the antagonistic supernatant method? What did you do to avoid clumps or empties? Did you adjust the final volumes? How did you avoid the absorbance by the culture dilutions?

How can I distinguish between mycelium interference with light and absorbance? I suppose the culture is not homogeneous ¿what is the principle to determinate the mycelial growth by spectrophotometry? What characteristics of the spectrophotometer has to allow you to determine fungal growth?

Furthermore, statural fungal cultures, sporulates on the interphase air/water which could be a problem. The other problem could be the secreted metabolites by the fungi.

Lines 217-219. Did you observe a reverse correlation for the added volume from highest to lowest with the slight increase in fungal growth? It would be better if you show the 48-well microtiter plate representative photos.

Line 261. I think the expression "data not shown" is not allowed in this journal. Please show the data.

Lines 274-276. Please provide the quantification of the dual culture assays. I also would like to see a correlation analysis with both data.

Lines 277-278. How many represent a large number for you? Considering this point what exactly you understand for HTP/HT?

Line 415. How were the plates scanned and processed?

Lines 423-424. Why the spore concentration varied for the different fungi? Please explain.

Line 433-435. I am not sure if the determination of fungal growth spectrophotometrically is the best and accurate and reliable way to determine fungal growth. I would like to see the determination of fungal growth though time and the spectrophotometric data in a graphic with its respective correlation analysis.

Please take a look at the following literature maybe you can find similarities with your methods. For me, this work just mixes and extend the described protocols.

<https://doi.org/10.1186/s13568-020-01132-1>

<https://doi.org/10.1101/611855>

<https://doi.org/10.1016/j.mimet.2021.106311>

Reviewer #2 (Comments for the Author):

Dear Author

The proposed work is very interesting and brings new approaches in the selection of beneficial microorganisms that can be used in the biological protection of plants against diseases caused by pathogenic fungi. Nevertheless, there is no empirical evidence supporting the advantages of the proposed methods over the current ones. The authors do not compare (numerically) the time-consumption, costs and effectiveness of the available methods with those proposed. Just minimizing the experience to a 48-well microtiter plate does not justify its high throughput. The authors do not analyze the biotest preparation time, its duration and the time it takes to perform observations and measurements. The authors also do not indicate the costs generated by this approach. This knowledge could be used to compare the proposed approaches with existing methods - therefore some of the results presented in the manuscript are rather discussion and conclusions are not substantiated.

I would like to point out that the resolution, visibility of some photos - Fig. 2 and Fig. S1 is very poor. The results cannot be properly assessed.

These and other comments are included in the PDF of the manuscript.

Staff Comments:

Preparing Revision Guidelines

Please return the manuscript within 60 days; if you cannot complete the modification within this time period, please contact me. If you do not wish to modify the manuscript and prefer to submit it to another journal, please notify me of your decision immediately so that the manuscript may be formally withdrawn from consideration by Microbiology Spectrum.

**Development of quantitative high-throughput screening methods for**
**identification of antifungal biocontrol strains**

Bodil Kjeldgaard^{1,2}, Ana Rute Neves^{1,3}, César Fonseca¹, Ákos T. Kovács², Patricia Domínguez- Cuevas^{1*}

¹Discovery, R&D, Chr. Hansen A/S, Denmark

²Bacterial Interactions and Evolution Group, DTU Bioengineering, Technical University of Denmark,
Denmark

³Current affiliation: Arla Foods Ingredients, Denmark

*For correspondence. E-mail: dkpacu@chr-hansen.com, Bøge Alle 10, 2970 Hørsholm, Denmark, Tel.
+4552180826

**Abstract**

Large screens of bacterial strain collections to identify potential biocontrol agents are often time
consuming, costly, and fail to provide quantitative results. In this study, we present two quantitative
and high-throughput methods to assess the inhibitory capacity of bacterial biocontrol candidates
against fungal phytopathogens. One method measures the inhibitory effect of bacterial culture
supernatant components on the fungal growth, while the other accounts for direct interaction between
growing bacteria and the fungus by co-cultivating the two organisms. The antagonistic supernatant
method quantifies the culture components' antifungal activity by calculating the cumulative impact of
supernatant addition relative to a non-treated fungal control, while the antagonistic co-cultivation
method identifies the minimal bacterial cell concentration required to inhibit fungal growth by co-

inoculating fungal spores with bacterial culture dilution series. Thereby, both methods provide
quantitative measures of biocontrol efficiency and allow prominent fungal inhibitors to be distinguished
from less effective strains. The combination of the two methods shed light on the type of inhibition
mechanisms and provide the basis for further mode of action studies. We demonstrate the efficacy of
the methods using *Bacillus spp.* with different levels of antifungal activities as model antagonists and
quantify their inhibitory potency against classic plant pathogens.

**Importance:** Fungal phytopathogens are responsible for tremendous agricultural losses on annual
basis. While microbial biocontrol agents represent a promising solution to the problem, there is a
growing need for high-throughput methods to evaluate and quantify inhibitory properties of new
potential biocontrol agents for agricultural application. In this study, we present two high-throughput
and quantitative fungal inhibition methods that are suitable for commercial biocontrol screening.

**Abbreviations**

Minimal inhibitory cell concentration, MICC; plant growth promoting rhizobacteria, PGPR; high-
throughput, HTP; potato dextrose agar, PDA, potato dextrose broth, PDB; cell-forming units, CFU's

**Keywords:** *Bacillus*, *Fusarium culmorum*, *Fusarium graminearum*, *Botrytis cinerea*, fungal growth
inhibition method, quantification of antifungal properties, biocontrol agents, bacterial-fungal co-
inoculation, bioactive compounds, biocontrol screening

**Introduction**

On annual basis, it is estimated that the global crop production suffers losses between 20 to 40 percent
due to pests and plant diseases [1]. Plant diseases alone are predicted to cost the global economy a
staggering \$220 billion per year [2]. Among other plant pathogens, fungal phytopathogens contribute
to considerable losses in agriculture and greatly impact food security in developing countries [3–5]. Not
only do fungal pathogens affect the yield, but fungal crop infections also lead to severe reductions of
post-harvest crop quality. For instance, accumulation of high levels of mycotoxins render crops unsafe
for human consumption and for animal forage [6,7]. In modern intensified agriculture, fungal diseases
are commonly fought using fungicides [8], but the rising fungicide resistance and chemical pollution
represent a challenge to the sustainable use of these chemicals in agriculture [9–12]. In addition, many
fungicides are hazardous to humans and may infer developmental toxicity, reproductive defects or
cancer [13,14]. Application of microbial biocontrol agents represent a safe alternative to the intensive
use of agrochemicals [11,15]. Biocontrol agents derived from plant growth promoting rhizobacteria
(PGPR) reside in close association with plant roots, and protect the plant from phytopathogens by
priming the plant defense response, competing for nutrients and/or directly antagonizing growth and
development of the pathogenic intruders [16,17]. Strains from *Pseudomonas*, *Burkholderia*,
*Streptomyces* and the *Bacillus* genera are well known for their antifungal capacity and for the
production of a large variety of bioactive metabolites [15,18–23]. Although, the inhibitory effect of
specific soil bacteria is well documented and recognized, there is a lack of quantitative and **HTP**
screening procedures to identify competent biocontrol agents. Consequently, many potential

biocontrol agents eventually fail to suppress plant diseases in field trials [24,25]. Classic antagonistic
screens assess the impact of the biocontrol candidates on the phytopathogen after co-inoculation on
solid media, which are referred to as dual-culture, plate confrontation or inhibition zone assays [24,26].
Such methods account for numerous factors including nutrients or space competition, cell surface
components and the induced or constitutive secretion of volatile or soluble metabolites [20,26–28].
Other antagonistic assays evaluate the effect of individual inhibitory components on the
phytopathogens' growth like volatiles, polyketides, lipopeptides, siderophores and lytic enzymes
including, chitinases, glucanases, and proteases [20,26]. More complex antagonistic assays, such as leaf
disc or seedlings assays [29,30], investigate the tripartite interaction between biocontrol candidate,
phytopathogen and plant host, while non-antagonistic assays assess the importance of complementary
inhibitory mechanisms including niche colonization and priming of the plant immune response [26].
Nevertheless, most screening systems are low throughput and provide only semi-quantitative
measurements of the inhibition potential against the fungus. Therefore, there is a need to develop
more efficient screening methods combining quantitative measurements of antimicrobial activity with
automation to increase speed and reduce resources required for the identification of good candidates.
Here, we describe two fungal inhibition methods to evaluate antifungal potency of potential biocontrol
agents. Both methods accommodate **high-throughput** (HTP) screening of bacterial biocontrol
candidates and provide accurate quantification of their inhibitory capacity. The major difference
between the two methods is represented by the use of growing bacterial cells as opposed to (cell-
inactive) culture supernatants. We demonstrate the efficacy of the methods using bacterial strains with
different antifungal performance. Using the two novel methods, antifungal properties of bacteria were

compared, and prominent fungal inhibitors were distinguished from less effective bacterial strains.
Both methods were developed utilizing *Fusarium culmorum* as model plant pathogen and *Bacillus* spp.
as model antagonists. In addition, the methods were further applied for screening *Bacillus* spp. against
other important phytopathogens, i.e. *Fusarium graminearum* and *Botrytis cinerea*, proving that our
method can readily be adjusted to other fungal species.

**Results**

**Co-inoculation of fungal spores with bacterial dilution series facilitates quantification of inhibition** 92 **potency**

The so-called dual-culture assay is among the most common screening methods to identify potent
fungal inhibitors from microbial collections [18,22,31–33]. Typically, the assay is performed by
inoculating potential biocontrol agents at a fixed distance from the pathogenic fungal inoculum on a
petri dish as illustrated in Fig. 1A. Subsequently, the biocontrol agent's ability to suppress fungal growth
is manually assessed by measuring the radius of the mycelial growth relative to the control or the size
of the inhibition zone [34–37]. However, accurate comparison and subsequent ranking of large
numbers of strains is difficult with this assay due to the format of the readout. To improve the
evaluation and accurate quantification of antifungal potency, we developed a HTP fungal inhibition
assay based on direct co-inoculation of bacterial cultures and fungal plant pathogenic spores. First,
bacterial overnight cultures were normalized to the same optical density at 600 nm (OD₆₀₀) and 5-fold
serial dilutions were prepared (down to an amount of approximately 3-5 bacterial cells inoculated). The
OD₆₀₀ of each strain used in the study was correlated to viable cell counts (colony forming units (CFUs)).

A fungal spore suspension was prepared and mixed thoroughly to ensure a homogeneous spore
distribution. Then, the bacterial dilution series were co-inoculated with a fixed quantity of fungal spores
to determine the minimal inhibitory cell concentration (MICC) that abolishes fungal growth (Fig. 1B).
With this setup, a low MICC indicates a higher antifungal potency for a given bacterial strain. The assay
was prepared on appropriate agar medium for fungal cultivation in 48-well microtiter plates and
incubated 5 days at room temperature before assessing the fungal growth.

Fig. 1: Fungal inhibition assays. (A) *B. subtilis* BCF001, *B. amyloliquefaciens* BCF007, *B. paralicheniformis*
BCF009 and *B. velezensis* BCF015 were spotted around central inocula of *F. culmorum*, *F. graminearum*
and *B. cinerea* on agar medium. Inhibition zones were observed 4-5 days post inoculation. (B) In each
well of a 48-well microtiter plate, molten PDA medium was aliquoted and let to solidify. 5-fold dilution
series of *Bacillus* strains were prepared from cultures normalized to the same OD₆₀₀. In each column,
consecutive wells were co-inoculated with the *Bacillus* dilution series and a fixed quantity of fungal
spores (constant volume of fungal spore suspension) on the agar surface. The spore suspensions were
mixed thoroughly before aliquoting. Assay results were evaluated by visual inspection after 5 days
incubation at room temperature.

Four *Bacillus* strains of different species were selected based on their differential properties for fungal
inhibition. The inhibitory capacity of the selected strains, *Bacillus subtilis* BCF001,
*Bacillus amyloliquefaciens* BCF007, *Bacillus paralicheniformis* BCF009 and *Bacillus velezensis* BCF015,
was quantified against *F. culmorum* DSM1094 using the developed method (Fig. 2B). The assay was
conducted as three biological independent replicates, rendering very similar results, which indicates
that the method is highly reproducible.

Fungal growth inhibition was scored as *no growth* (1), *weak growth* (2) and *(positive) growth* (3) (Fig.
2B). Fungal growth was defined by presence of visible hyphae in the well, including those that were
half-covered by fungal mycelia. No fungal growth was defined by complete absence of visible fungal
mycelia. Weak growth was defined by the presence of barely visible hyphae in all replicates, or by the

absence of growth in half of the replicates. The weak growth category included the wells where
 *F. culmorum* produced (orange) pigmentation, even in the absence of visible hyphae.

Fig. 2: Comparison of *Bacillus* spp. inhibitory properties against *F. culmorum*. (A) *F. culmorum* growth
 was scored following a three steps scale illustrated with three image examples per category. (B) Dilution
 series of *B. subtilis* BCF001, *B. amyloliquefaciens* BCF007, *B. paralicheniformis* BCF009 and *B. velezensis*
 BCF015 were prepared and inoculated in consecutive columns of a 48-well microtiter plate. A constant
 spore concentration of *F. culmorum* was inoculated in each well. ODs of serial dilutions are indicated
 on the left panel (OD₆₀₀).

Comparison of bacterial MICC allowed ranking of the strains in accordance to their fungal inhibition
properties (Fig. 3A, Table S1). The strains *B. amyloliquefaciens* BCF007 and *B. velezensis* BCF015 showed
the most potent inhibition properties, and even at the highest dilution step (corresponding to an initial
OD₆₀₀ of 6.4×10^{-6} , equivalent to 3-5 CFUs inoculated), the two strains were able to inhibit *F. culmorum*
growth. Determination of the minimal cell concentration that inhibits fungal growth even more
accurately would require to conduct the assay using serial dilutions at smaller dilution factors.
Nevertheless, the experimental setup is optimized for HTP screening of large strain collections.
*B. subtilis* BCF001 also displayed potent fungal inhibition properties, and abolished fungal growth up
to an OD₆₀₀ of 3.2×10^{-5} , corresponding to 19 CFU inoculated. *B. paralicheniformis* BCF009, however,
did not affect fungal growth even at the lowest dilution step, corresponding to an OD₆₀₀ of 4×10^{-3} ,
3.89×10^4 CFUs inoculated.

To generate visual and directly comparable plots of the inhibition results, we calculated a numerical
inhibition score that reflects the inhibitory capacity of each strain. In brief, the lowest cell concentration
that caused fungal growth inhibition was identified for each strain, and a numerical inhibition score was
calculated based on the natural logarithm to the MICC, by applying the empirical formula #1 (see
material and methods section) (Fig. 3A, Table S1). Plotting the inhibition scores clearly indicated that
*B. amyloliquefaciens* BCF007 and *B. velezensis* BCF015 were the most efficient strains inhibiting growth
of *F. culmorum*. *B. subtilis* BCF001 also showed a high inhibition score although smaller than the ones

calculated for the two former strains. Scoring results for *B. paralicheniformis* BCF009 matched the low
inhibitory activity of this strain against *F. culmorum*.

Fig. 3: Inhibition potency of *Bacilli* against fungal phytopathogens. The inhibition potency of *B. subtilis*
BCF001, *B. amyloliquefaciens* BCF007, *B. paralicheniformis* BCF009 and *B. velezensis* BCF015 was
assigned a numerical score based on the minimal inhibitory cell concentration (MICC) against (A)
*F. culmorum*, (B) *F. graminearum* and (C) *B. cinerea*. Inhibition scores were calculated by applying the
formula #1.

Evaluation of inhibition method with additional fungal phytopathogens

Using the newly developed method, the antifungal activity of *B. subtilis* BCF001, *B. amyloliquefaciens*
BCF007, *B. paralicheniformis* BCF009 and *B. velezensis* BCF015 was also quantified against other
phytopathogenic filamentous fungi, specifically against *F. graminearum* and *B. cinerea* strains. Images
acquired to record the experimental results, the calculated MICCs and inhibition scores can be found in
Fig. S1, Table S1. Interestingly, our inhibition assay proved applicable to these additional species of
plant pathogenic fungi. The initial spore concentration was the only parameter that required
adjustment when testing growth inhibition against the new fungal strains.

The previously described scoring system was applied to the results obtained for the three fungal species
(Fig. 3). While *B. amyloliquefaciens* BCF007 and *B. velezensis* BCF015 displayed the most efficient
growth inhibition of *F. culmorum*, differences between them and *B. subtilis* BCF001 were minimal when
assayed against *F. graminearum* DSM4528. Furthermore, the strain *B. subtilis* BCF001 proved more
efficient against *B. cinerea* Kern B2 compared to *B. amyloliquefaciens* BCF007 and *B. velezensis* BCF015.
Inhibition scoring results obtained for *B. paralicheniformis* BCF009 were in accordance with the poor
inhibitory properties of this strain, regardless of the fungal pathogen tested.

**Inoculation of fungal spores with bacterial culture supernatants validates the inhibitory importance** 190 **of supernatant components**

While the antagonistic co-inoculation method assesses several inhibitory mechanisms, the use of cell-
inactive supernatants accounts for the bioactivity of secreted metabolites, like enzymes, lipopeptides
and polyketides, on the fungus. In the antagonistic supernatant method, fungal growth was estimated
by OD₆₀₀ measurements in liquid cultures and therefore, reduced fungal growth in the presence of
bacterial metabolites reflects greater inhibition potency of a given strain. For this method, bacterial
cultures were grown overnight and normalized to the same OD₆₀₀ value prior to collection of the culture
supernatants by centrifugation (Fig. 4). A range of increasing supernatant volumes (10-80 µl) were
inoculated into fungal spore suspensions prepared in liquid medium in 48-well microtiter plates. To
avoid any possibility of misreads arising from remaining bacterial cells present in the supernatants,
bacteriostatic antibiotics were added to the fungal spore suspension to prevent bacterial growth. The

microtiter plates were incubated statically at 25°C in darkness. Subsequently, the fungal growth was
quantified by spectrophotometric measurement.

Fig. 4 Supernatant inhibition method. A fungal spore suspension was prepared with PDB medium and
the bacteriostatics tetracycline (TET) and chloramphenicol (CM). The suspension was mixed well and
aliquoted into each well of a 48-well microtiter plate. *Bacillus* overnight cultures were adjusted to 2
OD₆₀₀ and were subsequently centrifuged to collect the supernatants. In each column, *Bacillus*
supernatant volumes (10-80 µl) were added to consecutive wells. Fungal growth was evaluated by
spectrophotometric measurements after 5 days incubation at 25°C in darkness.

The inhibition assay demonstrated a reverse correlation between the added volume of bacterial cell-
inactive supernatant and growth of *F. culmorum*, *F. graminearum* or *B. cinerea* (Fig. 5A-C). For potent
fungal inhibitory strains, including *B. subtilis* BCF001, *B. amyloliquefaciens* BCF007 and *B. velezensis*
BCF015, increasing volumes of bacterial supernatant correlated with a progressive decrease of fungal
growth. Addition of *B. paralicheniformis* BCF009 supernatant impacted fungal growth to a much lesser
extent, which is in accordance with the results obtained with the co-inoculation method. In some cases,
a slight increase in fungal growth was observed in response to addition of the largest supernatant
volume (80 µl culture supernatant), suggesting an effect of supplemented nutrients (not consumed by
the bacteria), which outweighed the antifungal bioactive metabolites.

Fig. 5: Fungal Growth inhibition by *Bacillus spp.* culture supernatants. *Bacillus spp.* cultures were grown
 overnight in LB broth and normalized to OD₆₀₀ 2. Different volumes of cell-inactive supernatants (10,
 20, 40, 60 and 80 µl) were added to 48-well microtiter plates containing PDB medium and a fixed fungal
 spore concentration. Bacterial growth was inhibited by presence of bacteriostatic antibiotics in the
 culture medium (50 µg/ml chloramphenicol and tetracycline 10 µg/ml, respectively). Plates were
 incubated at 25°C without shaking for 60 h (in darkness) and fungal growth was measured by

spectrophotometry at 600 nm. Plots correspond to inhibition results for (A) *F. culmorum*, (B)
*F. graminearum* and (C) *B. cinerea*. Statistical significance for each supernatant volume compared to
the control was calculated based on biological triplicates. *P<0.05, **P<0.005, ***P<0.0005. For
quantification of the culture supernatant inhibition potency, an inhibition score was assigned to each
*Bacillus* strain by subtracting the accumulated relative fungal growth of (D) *F. culmorum*, (E)
*F. graminearum* and (F) *B. cinerea* in response to all supernatant volumes from the total potential
growth according to Equation #2. Standard deviations were calculated based on biological triplicates.

**Scoring of supernatant inhibition potency**

A numerical inhibition score was assigned to each strain by quantifying the fungal growth in response
to supernatant addition (5 volumes). The inhibition scores were calculated by subtracting the
accumulated relative fungal growth from total potential fungal growth in the 5 conditions tested
(Equation #2, see material and methods section).

The inhibition scores can be found in Table S1. Culture supernatants of *B. subtilis* BCF001 and
*B. amyloliquefaciens* BCF007 showed the most effective growth inhibition of *F. culmorum* (Fig. 5A and
D). Against *F. graminearum*, *B. subtilis* BCF001 culture supernatants proved most inhibitory, while
*B. subtilis* BCF001, *B. amyloliquefaciens* BCF007 and *B. velezensis* BCF015 all displayed similar inhibition
potency of *B. cinerea*. *B. paralicheniformis* BCF009 culture supernatants poorly inhibited *F. culmorum*
growth.

It is worth mentioning that assays prepared from different fungal spore stock solutions showed small
variations in final fungal growth (Fig. S2), possibly due to variations in the initial spore count or spore

viability. These variations could be minimized either by prolonging the incubation time until readout or
by preparing assays from a unique spore stock. For this reason, biological replicates were prepared with
the same fungal spore solution. Nevertheless, the overall conclusions remained unchanged despite
variations in the final fungal growth.

**Comparison of methods to remove or inactivate cells from culture supernatants**

Typically, bacterial cell-inactive supernatants are generated by filtration of the cell cultures [38–41]. In
contrast, in the proposed setup cell-inactive supernatants were generated by centrifugation to pellet
bacterial cells. To avoid cell growth of remaining bacteria in suspension, we added the bacteriostatic
antibiotics chloramphenicol and tetracycline. Plating of the fungal growth suspension on bacteria-
selective medium after the end point measurement produced no bacterial growth, indicating that the
antibiotics effectively inhibited growth of any putative remaining bacterial cells (data not shown). Final

[revised manuscript text omitted]

collections, 3) can provide valuable insights into type of inhibition mechanisms for further studies and,
4) allows easy comparison of strains by accurate quantification of their inhibition potency.

**Materials and Methods**

**Microbial species and growth conditions.** The fungal and bacterial strains used in the inhibition assays
are shown in Table 1. Fungal species were cultivated on potato dextrose agar (PDA) medium ([Carl
Roth]; potato infusion 4 g/l, glucose 20 g/l, agar 15 g/l, pH value 5.6 ± 0.2). For inhibition assays on PDA,
cultures were incubated at room temperature with natural light. For inhibition assays in broth, fungal
spores were inoculated in potato dextrose broth (PDB) medium ([Carl Roth]; potato infusion 6.5 g/l,
glucose 20 g/l, pH value 5.6 ± 0.2) and incubated at 25°C in darkness. *Bacilli* were grown overnight in
Lysogeny Broth ([Difco]; Bacto tryptone 10 g/l, [Oxoid]; yeast extract 5 g/l, [Merck]; NaCl 10 g/l, pH
value 7.2 ± 0.2) at 37°C with agitation at 250 rpm.

Table 1: Fungal and bacterial strains used in the present study.

Species and strains	Number	Source
Mold fungi		
F. culmorum	DSM1094	DSMZ German Collection

F. graminearum	DSM4528	DSMZ German Collection
B. cinerea	Kern B2	University of California, Davis
		
Bacteria		
B. subtilis	BCF001	Chr. Hansen A/S
B. amyloliquefaciens	BCF007	Chr. Hansen A/S
B. paralicheniformis	BCF009	Chr. Hansen A/S
B. velezensis	BCF015	Chr. Hansen A/S

**Fungal spore harvest.** Spores were harvested as described by Benoit et al., 2015 and Kjeldgaard et al.,
 2019. In brief, PDA plates were inoculated with fungi and incubated at 22°C with 16h light/8h dark
 cycles for at least 2 weeks. Fungal spores were harvested from PDA plates using saline tween solution
 (8 g/l NaCl, 0.05 ml/l Tween 80) and gentle scraping with an L-spreader. The spore solution was filtered
 through double layered Miracloth [Millipore] and pelleted by centrifugation at 5000 rpm for 10 min.
 The supernatant was discarded and the spore pellet resuspended in saline tween. Spore stocks were
 prepared by adding an appropriate concentration of glycerol.

**Quantification of fungal growth inhibition by co-inoculation with bacterial dilution series.** The fungal
 inhibition assay was prepared in 48-microtiter plates with 0.5 ml PDA in each well. *Bacillus* cultures
 were adjusted to 2×10^{-2} or 8×10^{-4} at OD₆₀₀ and 5-fold dilution series were prepared with 6 steps using
 peptone saline as diluent ([Millipore]; Maximum Recovery Diluent 9.5 g/L). 15 µl of each bacterial
 dilution was inoculated in consecutive wells by spotting the solution in the center of the wells. One
 strain was assigned per column. Following, a fungal spore solution was prepared with peptone saline
 and vortexed vigorously to disperse clumps of spores. 15 µl fungal spore solution was co-inoculated
 with the bacterial dilutions in each well. The approximate final fungal spore concentration of

*F. culmorum*, *F. graminearum* and *B. cinerea* was 5.5×10^4 spores/ml, 3.1×10^3 spores/ml and 7.5×10^5
spores/ml, respectively. Co-inoculated plates were sealed with 3M tape ([Millipore] 0.5 cm) to reduce
growth differences between inner and peripheric wells, and incubated at room temperature under
natural light conditions. After 5 days, the fungal inhibition was evaluated first by visual inspection and
later scored by scanning the plates. The minimal inhibitory cell concentration (MICC) was identified for
each strain and averaged between technical duplicates, then averaged between biological triplicates.
The MICC (CFU) against each fungal species was converted to an inhibition score by Equation #1 and
statistical significance was calculated by two-tailed student's T-test.

$$\#1 \quad \text{Inhibition score} = 5 - \frac{\ln(\text{MICC})}{2.5}$$

**Quantification of fungal inhibition potency by metabolites in bacterial (cell-inactive) supernatants.**

The fungal inhibition assay was prepared in 48-microtiter plates with 0.5 ml PDB medium. Spores of
*F. culmorum*, *F. graminearum* or *B. cinerea* were added to a final concentration of 1.1×10^4 spores/ml,
6.1×10^2 spores/ml and 7.5×10^4 spores/ml respectively. The spore suspensions were mixed thoroughly
to ensure a homogeneous distribution. *Bacillus* spp. cultures were adjusted to OD₆₀₀ of 2 and
centrifuged to pellet the cells. The supernatants were collected and the cell pellets discarded. Increasing
bacterial supernatant volumes (ranging from 10-80 μ l) were inoculated into the fungal spore
suspension. Two approaches were implemented for sterilization of the supernatant to omit bacterial
growth. Either the bacteriostatic antibiotics chloramphenicol and tetracycline were added to the fungal
spore suspension to a final concentration of 50 μ g/ml and 10 μ g/ml, respectively, or the supernatants
were sterilized by filtration ([Sartorius] Minisart Syringe Filter 0.2 μ m) prior to co-inoculation with the

fungal spores. The plates were sealed with 3M tape and incubated at 25°C without shaking in the
darkness. The fungal growth was quantified by spectrophotometric measurements (OD₆₀₀) after 48h
for *F. graminearum* and *B. cinerea* inhibition assays and after 67-72h for *F. culmorum* inhibition assays.
For more accurate fungal growth estimation, the OD₆₀₀ was measured using a 5x5 well-scanning matrix.
Following the evaluation of fungal growth, the supernatants were collected and plated on LB with
fungicides (50 mg/L nystatin) to check for unwanted bacterial growth. Statistical analyses were done to
compare to the effect of 1) bacterial culture supernatants on fungal growth, 2) filtration and antibiotics
on the bacterial supernatants potency, and 3) prokaryotic antibiotics on fungal growth. Statistical
significance was calculated by students T-test (two-tailed). A numerical inhibition score was assigned
to each strain by quantifying the fungal growth in response to supernatant addition (5 volumes). The
inhibition scores were calculated by subtracting the accumulated relative fungal growth from total
potential fungal growth (5 conditions tested) as shown in Equation #2:

$$444 \quad \#2 \quad 5 - \text{SUM} (\text{relative fungal growth})$$

**References**

- [1] Food and Agriculture Organization of the United Nations. Plant health and food security. Int
Plant Prot Conv 2017.
- [2] FAO. New standards to curb the global spread of plant pests and diseases 2019.
<http://www.fao.org/news/story/en/item/1187738/icode/> (accessed March 28, 2020).
- [3] Savary S, Ficke A, Aubertot JN, Hollier C. Crop losses due to diseases and their implications for
global food production losses and food security. Food Secur 2012;4:519–37.

- <https://doi.org/10.1007/s12571-012-0200-5>.
- [4] Almeida F, Rodrigues ML, Coelho C. The Still Underestimated Problem of Fungal Diseases
Worldwide. *Front Microbiol* 2019;10:214. <https://doi.org/10.3389/fmicb.2019.00214>.
- [5] Fisher MC, Henk DA, Briggs CJ, Brownstein JS, Madoff LC, Mccraw SL, et al. Emerging fungal
threats to animal, plant and ecosystem health. *Nature* 2012;484.
<https://doi.org/10.1038/nature10947>.
- [6] Hoffmann V, Jones K, Leroy JL. The impact of reducing dietary aflatoxin exposure on child linear
growth: A cluster randomised controlled trial in Kenya. *BMJ Glob Heal* 2018;3:e000983.
<https://doi.org/10.1136/bmjgh-2018-000983>.
- [7] De Lucca AJ. Harmful fungi in both agriculture and medicine. *Rev Iberoam Micol* 2007;24:3–13.
[https://doi.org/10.1016/s1130-1406\(07\)70002-5](https://doi.org/10.1016/s1130-1406(07)70002-5).
- [8] Lucas JA, Hawkins NJ, Fraaije BA. The Evolution of Fungicide Resistance. *Adv Appl Microbiol*
2015;90:29–92. <https://doi.org/10.1016/bs.aambs.2014.09.001>.
- [9] Hahn M. The rising threat of fungicide resistance in plant pathogenic fungi: Botrytis as a case
study. *J Chem Biol* 2014;7:133–41. <https://doi.org/10.1007/s12154-014-0113-1>.
- [10] Hellin P, King R, Urban M, Hammond-Kosack KE, Legrève A. The adaptation of *Fusarium*
*culmorum* to DMI Fungicides Is Mediated by Major Transcriptome Modifications in Response to
Azole Fungicide, Including the Overexpression of a PDR Transporter (FcABC1). *Front Microbiol*
2018;9:1385. <https://doi.org/10.3389/fmicb.2018.01385>.
- [11] Brauer VS, Rezende CP, Pessoni AM, De Paula RG, Rangappa KS, Nayaka SC, et al. Antifungal
agents in agriculture: Friends and foes of public health. *Biomolecules* 2019;9:521.

<https://doi.org/10.3390/biom9100521>.

[12] Zubrod JP, Bundschuh M, Arts G, Brühl CA, Imfeld G, Knäbel A, et al. Fungicides: An Overlooked
Pesticide Class? *Environ Sci Technol* 2019;53:3347–65.

<https://doi.org/10.1021/acs.est.8b04392>.

[13] Gupta PK. Herbicides and Fungicides. *Reprod. Dev. Toxicol.*, 2011, p. 503–21.

<https://doi.org/10.1016/B978-0-12-382032-7.10039-6>.

[14] Kim KH, Kabir E, Jahan SA. Exposure to pesticides and the associated human health effects. *Sci*
*Total Environ* 2017;575:525–35. <https://doi.org/10.1016/j.scitotenv.2016.09.009>.

[15] Cawoy H, Bettiol W, Fickers P, Ongena M. Bacillus-Based Biological Control of Plant Diseases.
*Pestic Mod World - Pestic Use Manag* 2011;1849:273–303. [https://doi.org/DOI:

[revised manuscript text omitted]

**Supplementary**

Table S1: The minimal inhibitory cell concentration (MICC) was determined for *B. subtilis* BCF001,
 *B. amyloliquefaciens* BCF007, *B. velezensis* BCF015 and *B. paralicheniformis* BCF009, as the lower
 number of cells (CFUs) that inhibit growth of *F. culmorum*, *F. graminearum* and *B. cinerea* by co-
 cultivation. The MICC against each fungal species was converted to an inhibition score by formula #1.
 For quantification of the culture supernatant inhibition potency, an inhibition score was calculated by
 formula #2 for each bacterial strain and for each experimental cell-inactivation method.

	BCF001	BCF007	BCF009	BCF015
MICC (CFU)				
F. culmorum	19	<3	>3.89*10 ⁴	<5
F. graminearum	57	60	>1.09*10 ⁵	74
B. cinerea	67	150	>1.10*10 ⁵	153
Co-cultivation inhibition score				
F. culmorum	3.8	4.5	0.8	4.4
F. graminearum	3.4	3.4	0.4	3.3
B. cinerea	3.3	3.0	0.4	3.0
Supernatant inhibition score (antibiotics)				
F. culmorum	2.63	2.81	0.62	2.06
F. graminearum	2.14	1.57	-0.43	1.81
B. cinerea	3.53	3.43	1.75	3.50
Supernatant inhibition score (filtration)				
F. culmorum	2.52	2.64	0.46	2.13
F. graminearum	2.04	1.22	-0.50	1.81
B. cinerea	3.55	3.42	1.32	3.53
Supernatant inhibition score (filtration and antibiotics)				
F. culmorum	2.47	2.56	0.26	1.81
F. graminearum	2.35	1.37	-0.40	1.72
B. cinerea	3.50	3.35	0.81	3.50

Fig. S1: Fungal inhibition assay with *Bacillus* dilution series. Dilution series of *B. subtilis* BCF001,
 *B. amyloliquefaciens* BCF007, *B. paralicheniformis* BCF009 and *B. velezensis* BCF015 were prepared and
 inoculated in consecutive columns of a 48-well microtiter plate. In each well, a constant spore
 concentration was inoculated of (A) *F. graminearum* and (B) *B. cinerea*. Pictures are representative of
 triplicates.

Fig. S2: Variation of *F. culmorum* growth with *Bacillus* supernatants. Inhibition assays with *F. culmorum*
 were prepared from different spore solutions and reached different final growth after 67h (A) and 72h
 (B) cultivation in PDB medium both with and without bacterial culture supernatants. Growth was

evaluated by spectrophotometric measurements (OD₆₀₀). Standard deviations were calculated from
 biological triplicates.

Fig. S3: Growth of *F. culmorum*, *F. graminearum* and *B. cinerea* without and without addition of
 antibiotics (50 µg/ml chloramphenicol and 10 µg/ml tetracycline). ***P<0.0005

Table S2: Statistical significance of different *Bacillus* supernatant sterilization methods on fungal
 inhibition results. Statistical significance was calculated using student's t-test. Statistical significant
 values (p<0.05) are indicated in red.

Filtered supernatants with antibiotics compared to supernatants with antibiotics						
Supernatant volume (µl)		10	20	40	60	80
F. culmorum	BCF001	0.43	0.55	0.69	0.18	0.12
	BCF007	0.21	0.01	0.24	0.82	0.11
	BFC009	0.97	0.2	0.05	0.08	0.01
	BFC015	0.77	0.31	0.09	0.01	0.04
F. graminearum	BCF001	0.88	0.09	0.01	0	0.94
	BCF007	0.31	0.14	0.66	0.85	0.06
	BFC009	0.3	0.44	0.45	0.61	0.78
	BFC015	0.28	0.01	0.6	0.34	0.32

B. cinerea	BCF001	0.11	0.61	0.4	0.89	0.71
	BCF007	0.81	0.51	0.08	0.34	0.34
	BFC009	0.07	0.1	0.02	0.1	0.22
	BFC015	0.46	0.51	0.06	0.82	1

Filtered supernatants with antibiotics compared to filtered supernatants

Supernatant volume (μl)		10	20	40	60	80
F. culmorum	BCF001	0.77	0.04	0.97	0.52	0.84
	BCF007	0.02	0	0.06	0.02	0.87
	BFC009	0.07	0.13	0.62	0.97	0.09
	BFC015	0.42	0	0.18	0.09	0.04
F. graminearum	BCF001	0.39	0.2	0.01	0.03	0.32
	BCF007	0.42	0.75	0.08	0.05	0.98
	BFC009	0.29	0.8	0.04	0.91	0.37
	BFC015	0.09	0.93	0.71	0.63	0.72
B. cinerea	BCF001	0.55	0.71	0.52	0.35	0.27
	BCF007	0.46	0.82	0	0.11	0
	BFC009	0.24	0.01	0.47	0.22	0.33
	BFC015	0.15	0.89	0.03	0.35	0

Filtered supernatants compared to supernatants with antibiotics

Supernatant volume (μl)		10	20	40	60	80
F. culmorum	BCF001	0.38	0.29	0.7	0.04	0.16
	BCF007	0.31	0	0.23	0.01	0.41
	BFC009	0	0.71	0.07	0.07	0.13
	BFC015	0.58	0.61	0.47	0.69	0.27
F. graminearum	BCF001	0.42	0.82	0.16	0.81	0.37
	BCF007	0.19	0.12	0.18	0.03	0.07
	BFC009	0.19	0.54	0.14	0.42	0.24
	BFC015	0.2	0.19	0.53	0.81	0.16
B. cinerea	BCF001	0.2	0.32	0.35	0.5	0.44
	BCF007	0.03	0.47	0.04	0.36	0
	BFC009	0.29	0.47	0.06	0.13	0.56
	BFC015	0.32	0.7	0.06	0.19	0.01

Fig. S4: Comparison of methods to circumvent bacterial cell growth in fungal inhibition assays with
 bacterial supernatants. Fungal spore suspensions of *F. culmorum* (A,B), *F. graminearum* (C,D) and
 *B. cinerea* (E,F) were inoculated with filtered bacterial culture with antibiotics (A, C, and E) or without
 addition of antibiotics (B, D, and F). *P<0.05, **P<0.005, ***P<0.0005.

Fig. S5: Comparison of supernatant inhibition potencies using *Bacillus* cell inactivation by filtration and
 antibiotic addition or by filtration only. The culture supernatant inhibition potencies of *B. subtilis*
 BCF001, *B. amyloliquefaciens* BCF007, *B. paralicheniformis* BCF009 and *B. velezensis* BCF015 against
 *F. culmorum* (A,D), *F. graminearum* (B,E) and *B. cinerea* (C,F) were calculated by applying the formula
 #2. The effect of bacterial cell inactivation by filtration and antibiotic addition (A-C) was compared to
 bacterial cell inactivation by filtration (D-F).

Reviewer comments

Reviewer #1 (Comments for the Author):

The manuscript describes two quantitative and high-throughput methods to evaluate the inhibitory capacity of bacterial biocontrol candidates against fungal phytopathogens. The first described method uses growing bacteria cocultured with the fungus, whereas the second one shows potential to measure the inhibitory effect of bacterial culture supernatant components on the fungal growth.

Since my view, the manuscript is well written and technically sounds, however, I have some major and minor concerns depicted below:

Lines 1-2. Since my view, the title must reflect that the methods were designed for the screening of biocontrol bacterial strains.

- The authors agree with the reviewer and therefore the title has been modified to: *Quantitative high-throughput screening methods designed for identification of bacterial biocontrol strains with antifungal properties*

Line 25 and subsequently. Please write down ".spp" in plane letter instead of italic.

- .spp is corrected to regular font. Thank you for commenting.

Line 55. I am not sure if this part of the sentence is right. I read the sentence as if the biocontrol agents emerge/born from PGPR.

- Text change line 54: *Biocontrol agents reside in close association with plant surfaces i.e. leaves or roots, and protect the plant from phytopathogens by priming the plant defense response, competing for nutrients and/or directly antagonizing growth and development of the pathogenic intruders*

Line 61. What HTP means? Please describe every abbreviature or acronym you use through the text. Furthermore, I think the right abbreviation for high-throughput is "HT". Please revise it.

- High-throughput has now been abbreviated to HT and high-throughput screening to HTS, respectively.

Line 79. Why do you think your proposed method belongs to high-throughput approaches? I do not agree with such statement. To my knowledge HT approaches are designed for testing from hundreds to thousands of samples in a short period of time.

- These screening methods have subsequently been used in our department to evaluate the antifungal inhibition potency of 20000+ strains within a short time frame. For instance, the antagonistic co-inoculation assay accommodates approximately 200-250 strains per day when using a Hamilton liquid handling robot. The assay allow screening of approximately 800-1000 strains per week. The supernatant inhibition assay accommodates 150-200 strains per day allowing a weekly throughput of approximately 1000 strains with aid of a robotic process workflow/automation.

- In the manuscript, we chose to demonstrate the functionality of the assays with a few selected strains for simplicity. Consequently, a paragraph has been added to the manuscript about the throughput of the assays (see Line 78 and 244).

Lines 100-101. Spores and mycelium do not have the same structure and composition of their cell walls, so, why not to include mycelium?

- We chose to use fungal spores to include the impact of *Bacilli* on fungal spore germination in the assessment, in addition to the impact on fungal mycelial growth. In addition, normalization of fungal spore inoculum is much easier than normalization of mycelium inoculum size, which is challenging. Using mycelial inoculum could decrease the reproducibility and accuracy of the assays
- In order to clarify we have introduced a text change in line 100: *Fungal spores rather than mycelium were used as initial inoculum in the assay to allow assessment of biocontrol agents impact on spore germination in addition to fungal growth.*

Figure 1A. Contrasting with the first two plates, I cannot see the fungus nor the bacteria in the third plate. Why?

- The backside of the *B. cinerea* plate is not shown. This is now noted in the figure legend (Line 634). This fungus does not produce orange pigments typically found accumulated by *Fusarium* species, which are in some cases more evident from the backside of the plate.

Figure 2A. What should I see in each column? I can see different morphologies and growth, then, I do not know why you said that there is not fungal growth. Please explain. Why did you not include the control of the fungi and the bacteria growing alone? That could help to understand what you meant. I can see fungal growth in all nine wells with different morphology, so why you stated that there is no growth in the first line. Should I see something that you did not explain?

- Figure 2A is meant to illustrate how the fungal growth or growth reduction in response to *Bacilli* was scored. The top row corresponds to bacterial growth only without clear presence of fungus. Selected pictures have been exchanged to ensure a clearer illustration.

Figure 2B. If I apply a zoom on the different wells, I can see growth on the different lines, so I would like to see the control of the fungi and the different Bacilli growing alone under the different conditions.

- Fungal growth in the absence of bacteria and bacterial growth in the absence of fungus have been added to figure 2B and figure legend modified accordingly (lines 643 and 649).

Lines 145-148. I cannot see in figure 2 such dilutions and does not seem to be the initial OD as stated. Have I got confused?

- The OD dilution series indicated next to figure 2B has been corrected. Thank you for noticing the discrepancy between the figure and the text.

Line 152. Did you mean 3.2×10^{-6} .

- The CFU's and OD's have been corrected and is in accordance with figure 2 now. Thank you for noticing.

Figure 3A. Above, you concluded that *B. amyloliquefaciens* and *B. velezensis* showed the most potent inhibition properties, and even at the highest dilution step against *F. culmorum*. However, *B. subtilis* showed similar results. It is necessary an statistical analysis.

- The significance of MICCs from each strain against each fungus was calculated and added to supplementary data:
 - A sentence was added in the text referring to the supplementary table with calculated MICC significance by comparison of the strain with students t-test (Table S2).
 - In addition, a sentence was added to explain that similarly potent strains can be distinguished by adjusted the spectrum of the dilution series. Zoom in on the effective MICC so to speak (see line 133 and line 166).

Lines 191-202. How did you determine the fungal growth by OD in the antagonistic supernatant method? What did you do to avoid clumps or empties? Did you adjust the final volumes? How did you avoid the absorbance by the culture dilutions? How can I distinguish between mycelium interference with light and absorbance? I suppose the culture is not homogeneous ¿what is the principle to determinate the mycelial growth by spectrophotometry? What characteristics of the spectrophotometer has to allow you to determine fungal growth? Furthermore, statical fungal cultures, sporulates on the interphase air/water which could be a problem. The other problem could be the secreted metabolites by the fungi.

Below we add a point-by-point answer to the reviewer's concerns:

- Spectrophotometry was used to determine the fungal growth as previous studies show good correlation between optical density measurements and fungal weight (Banerjee et al., 1993; Broekaert et al., 1990; Langvad, 1999; Petrikou et al., 2001). A sentence was added referring to the previous studies (see line 183).
- The fungal growth medium (potato dextrose broth) shares almost the same absorbance as the bacterial growth medium (Lysogeny Broth) and so the addition of supernatant dilutions should not affect the OD of the clear medium. The final volumes were however not adjusted, but a reverse correlation between supernatant addition and fungal growth was nevertheless observed.
- During assay preparation, the fungal spore solution was mixed thoroughly by vortex to ensure a homogeneous distribution and avoid clumping of spores.
- To account for empty space and clumps an area scanning protocol was used for measuring the fungal growth in each well (25 measurements/well), as we noticed that a single point measurement was not adequate. The measures were averaged within each well. The procedure is described in the main text for clarification and materials and methods (see lines 183 and 411).
- An additional figure (Figure S2) has been added to supplemental showing the OD measurements throughout a fungal growth curve.
- The fungal growth was evaluated before the sporulation and pigment production stages were reached, avoiding that they could potentially interfere with the optical density measurements (within 2-3 days cultivation).

Lines 217-219. Did you observe a reverse correlation for the added volume from highest to lowest with the slight increase in fungal growth? It would be better if you show the 48-well microtiter plate representative photos.

- A reverse correlation was observed between the added volumes (from lowest to highest) and the fungal growth (progressively decreasing). In some cases, the largest added volume (80ul) resulted in a slight increase in fungal growth in respect to the immediate lower added volume. This is probably related to the positive effect that higher content of nutrients added have on the final

fungal growth that slightly counteracts the inhibitory effect of bioactive metabolites. Adding larger volumes from this point on would not bring the fungal growth inhibition effect further. A sentence was added in the manuscript to explain this phenomenon (see line 195).

- The fungal growth in liquid is less clear on pictures as the cultivation time (2-3 days) does not allow for development of arial hyphae or pigmentation as observed in the antagonistic co-cultivation assay that was incubated for a longer time (5 days) to allow distinction between the two species. We consider that the pictures of the liquid cultures would not be informative, contrary to the optical density values that clearly show the fungal growth decrease in response to the bacterial supernatants' addition.

Line 261. I think the expression "data not shown" is not allowed in this journal. Please show the data.

- The data that the reviewer requests correspond to a picture of an empty cultivation plate with selection for bacterial growth (LB agar with nystatin). We estimated that this picture would not more informative than the sentence. To avoid the term "data not shown" we have rephrased the sentence to indicate that there was absence of bacterial growth (line 221).

Lines 274-276. Please provide the quantification of the dual culture assays. I also would like to see a correlation analysis with both data.

- We consider dual-culture assays to be semi-quantitative methods at best. Factors like the moisture of the agar plates or even the distance between the inoculation points of the biocontrol agent and the pathogen would bias the relative inhibition and lack reliability. Therefore, although dual-culture assays are simple and allow for a fast evaluation of biocontrol agents potential, the authors did not attempt a direct comparison of the methods.
- To respond to the reviewers request we have analyzed data from the dual-culture assay plates, calculated the corresponding inhibition scores and performed statistical analysis (see below).
- **Quantification of fungal growth inhibition by classical dual culture assay.** A volume of 10 μ l fungal spore solution was inoculated in the center of a PDA plate. *F. culmorum*, *F. graminearum* and *B. cinerea* were inoculated in final concentrations of $1.1 \cdot 10^7$ spores/ml, $1.25 \cdot 10^5$ spores/ml and $2 \cdot 10^7$ spores/ml, respectively. In a perimeter surrounding the central fungal inoculum, 10 μ l overnight culture were inoculated of the respective *Bacillus* cultures adjusted to OD₆₀₀ 0.2. The plates were sealed with 3M tape ([Millipore] 0.5 cm) and incubated at room temperature for 6 days. The fungal growth was evaluated in ImageJ (Fiji).

Figure 1. Percental growth reduction in response to confrontation with BCF001, BCF007, BCF009 and BCF015 of the fungi (A) *F. culmorum*, (B) *F. graminearum*, and (C) *B. cinerea*. The fungal control shows growth reduction (0%) in the absence of bacteria. Statical significant values from the respective fungal controls are indicated by *P<0.05, **P<0.005, ***P<0.0005.

Table 1. The growth reduction in response to BCF001, BCF007, BCF009 and BCF015 for each fungus was compared between strains. Statistically significant values ($p < 0.05$) are indicated in red.

Compared strains	F. culmorum	F. graminearum	B. cinerea
BCF001/BCF007	0.01747	0.56297	0.02335
BCF001/BCF009	0.00022	0.00168	0.00000
BCF001/BCF015	0.04118	0.92792	0.14289
BCF007/BCF009	0.00001	0.00077	0.00002
BCF007/BCF015	0.71780	0.46842	0.18024
BCF009/BCF015	0.00004	0.00066	0.00001

Lines 277-278. How many represent a large number for you? Considering this point what exactly you understand for HTP/HT?

- As commented previously, these assays were afterwards used for evaluation of antifungal potential of 20000+ strains within a short time frame. For instance, the antagonistic co-inoculation assay accommodates 250 strains per day when using a Hamilton liquid handling robot allowing screening of 800-1000 strains per week. In the manuscript, we aim to demonstrate the functionality of the assay with a few selected strains for simplicity.

Line 415. How were the plates scanned and processed?

Fungal inhibition by the antagonistic co-inoculation assay was evaluated by visual inspection of the plates and the plates were imaged by scanning the top of the plates using an Epson Perfection V800 Photo device. Line 388-389. Text was modified to clarify how plates were scanned and processed.

Lines 423-424. Why the spore concentration varied for the different fungi? Please explain.

- The fungal stocks may not have the same germination efficiency. Hence, the spore concentration was adjusted for each species to allow visualization of bacterial supernatant impact. In addition, different fungal species may be more or less susceptible to bacterial metabolites (enzymes, lipopeptides and polyketides). Therefore, the fungal spore concentration was adjusted, also to enable visualization of bacterial supernatant impact and allow comparison of strains' potency. For each fungus, it is necessary to optimize the initial spore concentration that allow visualization of the dynamic dilution range of bacterial cultures.

Line 433-435. I am not sure if the determination of fungal growth spectrophotometrically is the best and accurate and reliable way to determine fungal growth. I would like to see the determination of fungal growth though time and the spectrophotometric data in a graphic with its respective correlation analysis.

- As stated previously fungal growth curves were added to the supplemental materials (see Figure S2).

Please take a look at the following literature maybe you can find similarities with your methods. For me, this work just mixes and extend the described protocols.

<https://doi.org/10.1186/s13568-020-01132-1>

<https://doi.org/10.1101/611855>

<https://doi.org/10.1016/j.mimet.2021.106311>

- We appreciate these suggestions for literature references and recognize that our methods build upon previous methods. However, we argue that the antagonistic assays presented in our study provide quantification of fungal inhibition potency not offered by other HT methods. The measuring of inhibition zones as presented in study <https://doi.org/10.1186/s13568-020-01132-1> and the study <https://doi.org/10.1101/611855> is tedious and arguably quantitative. Distinction between similar strains is difficult and results provide limited accuracy of inhibition potency determination. We argue that that identification of the minimal effective cell concentration and the scoring of supernatant inhibition capacity allow accurate comparison of strains' potency. In classic antagonistic assays that measure inhibition zones, factors such as inoculum size, diffusion rate of metabolites, and differential conditions between agar plates contribute to the inaccuracy and impairs ranking of biocontrol candidates. Both <https://doi.org/10.1186/s13568-020-01132-1> and <https://doi.org/10.1101/611855> were cited in the discussion line 283. In addition, the study <https://doi.org/10.1186/s13568-020-01132-1> was cited in line 313 and line 335.
- The study <https://doi.org/10.1016/j.mimet.2021.106311> indeed presents a method for screening 96 bacterial isolates for antifungal activity at the same time. However, the method lacks quantification of antifungal properties for comparison of strain's potency and not achievable for testing certain fungal strains. The method is valid to confirm antifungal activity but does not allow ranking of strains' potencies.

Reviewer #2 (Comments for the Author):

Dear Author

The proposed work is very interesting and brings new approaches in the selection of beneficial microorganisms that can be used in the biological protection of plants against diseases caused by pathogenic fungi. Nevertheless, there is no empirical evidence supporting the advantages of the proposed methods over the current ones. The authors do not compare (numerically) the time-consumption, costs and effectiveness of the available methods with those proposed. Just minimizing the experience to a 48-well microtiter plate does not justify its high throughput. The authors do not analyze the biotest preparation time, its duration and the time it takes to perform observations and measurements. The authors also do not indicate the costs generated by this approach. This knowledge could be used to compare the proposed approaches with existing methods - therefore some of the results presented in the manuscript are rather discussion and conclusions are not substantiated. I would like to point out that the resolution, visibility of some photos - Fig. 2 and Fig. S1 is very poor. The results cannot be properly assessed.

- The throughput and estimation of time-consumption are described above. A paragraph was added to the manuscript (see line 243-249).
- All figures are supplied separately from the manuscript in better quality than the pdf file

Other comments included in the PDF of the manuscript:

Line 271: I agree that numerical methods are, as the name suggests, more measurable. In the case of bi-culture on-plate bioassay, the antagonist-pathogen interactions can also be represented numerically, both by measuring growth, and then the percentage of growth reduction, as well as by a multistage scale of overgrowth of the plate surface. In this chapter the comparison of the two approaches is more of a discussion than presenting evidence: comparison of time-consumption, costs, etc.

- Indeed, growth reduction can be measured on classical dual culture assays. However, such measures are manually laborious and arguably quantitative. Differentiation between strains with similar inhibition potency is difficult. The answer to the reviewer's comment, quantification of

inhibition potency from the dual-culture assay has been included above as part of the answers to reviewer 1 (page 4 of the response to reviewers).

- Time estimation and throughput were added to the section comparing the methods used in the study (see lines 243-247).

Line 411: Please complete the method by which the spore concentration was determined.

- A sentence was added to describe the procedure for determination of spore concentration (see line 374).

February 7, 2022

Dr. Patricia Domínguez-Cuevas
Chr. Hansen
Discovery, R&D
Bøge Allé 10-12
Hørsholm 2970
Denmark

Re: Spectrum01433-21R1 (Quantitative high-throughput screening methods designed for identification of bacterial biocontrol strains with antifungal properties)

Dear Dr. Patricia Domínguez-Cuevas:

I'm pleased to inform you that your manuscript has been deemed suitable for publication in Microbiology Spectrum.

Your manuscript has been accepted, and I am forwarding it to the ASM Journals Department for publication. You will be notified when your proofs are ready to be viewed.

Sincerely,

Lea Atanasova
Editor, Microbiology Spectrum